# Measuring and Mitigating Rapport Bias of Large Language Models under Multi-Agent Social Interactions

**Maojia Song**[*]
Singapore University of Technology and Design (SUTD)

**Tej Deep Pala**[*]
Nanyang Technological University (NTU)

**Ruiwen Zhou**
National University of Singapore (NUS)

**Weisheng Jin**
Singapore University of Technology and Design (SUTD)

**Amir Zadeh, Chuan Li**
Lambda Labs

**Dorien Herremans**
Singapore University of Technology and Design (SUTD)

**Soujanya Poria**[†]
Nanyang Technological University (NTU)

## Abstract

Large language models (LLMs) are increasingly integrated into multi-agent systems (MAS), where peer interactions shape individual decisions. While prior work has mainly examined conformity bias, we broaden the view to include how LLMs build rapport from prior interactions, discern and integrate high-quality peer information, and resist misleading inputs—abilities essential for achieving collective intelligence under complex social dynamics. We introduce KAIROS, a benchmark that simulates quiz-style collaboration with peer agents whose rapport levels and behaviors can be precisely controlled in both historical interactions and the current round. This unified setup enables systematic analysis of how rapport, peer actions, and the model's self-confidence jointly influence decision-making. Using KAIROS, we evaluate prompting, supervised fine-tuning, and reinforcement learning via Group Relative Policy Optimization (GRPO). Results show that model scale is a primary factor moderating susceptibility to social influence: larger models are more resilient and benefit from prompting-based mitigation, whereas smaller models remain vulnerable. Only carefully configured GRPO training yields consistent robustness and performance gains for small models. Our code and datasets are available at: https://anonymous.4open.science/r/KAIROS-4F71.

## 1 Introduction

Large Language Models (LLMs) are increasingly integrated into multi-agent systems (MAS), where they must interact, reason, and collaborate with other agents Chen et al. (2024b); Tran et al. (2025). However, like humans, LLMs are vulnerable to social and cognitive biases such as conformity, overconfidence, and herd behaviour Piatti et al. (2024); Weng et al. (2025); Yan et al. (2025). When exposed to peer responses, LLMs may adjust their outputs not only to align with group consensus but also due to misplaced trust in unreliable agents Cho et al. (2025). These tendencies pose a critical challenge in collective decision-making, where a single flawed response can propagate across agents, cascading through the system, and ultimately compromise the reliability of the entire multi-agent framework.

While prior studies have examined conformity in controlled, isolated settings Weng et al. (2025); Zhu et al. (2024), the field still lacks a comprehensive framework for simulating interactive social

---

[*]Equal contribution.
[†]Corresponding author: soujanya.poria@ntu.edu.sg.

environments and systematically evaluating LLM behaviour under varying conditions of rapport, current peer behaviour, and self-confidence. To fill this gap, we introduce KAIROS, a benchmark designed to assess LLMs in socially grounded, multi-agent scenarios. It simulates quiz-style multiple-choice contests in which the model engages with peer agents of varying reliability, leveraging both historical rapport and its own confidence to make decisions. Model behaviour is evaluated using four metrics: **accuracy**, the overall task success rate; **utility**, the ability to *correct its own errors* with peer input; **resistance**, the ability to *maintain correct judgments* despite misleading peers; and **robustness**, the change in accuracy from the original to the socially influenced setting, capturing stability under social interaction.

Beyond measuring susceptibility to social cues, our aim is to identify mitigation strategies that enhance model performance in multi-agent social simulations. We investigate three approaches: prompting, supervised fine-tuning, and reinforcement learning via GRPO. For GRPO, we systematically vary four factors: system prompt design, reward structure, incorporation of multi-agent context, and data filtering strategies. Our results show that GRPO, when trained with carefully configured MAS signals and outcome-based rewards, substantially outperforms prompting and SFT baselines, improving core task performance while preserving robustness against social perturbations. In contrast, alternative methods, though sometimes improving surface-level accuracy, often fail to generalise across protocol shifts, revealing persistent weaknesses in social reasoning. These findings highlight that accuracy alone is insufficient; resilient reasoning under social interference remains a central challenge for multi-agent generalisation.

Our work provides a systematic evaluation of LLM performance in socially interactive, trust-sensitive settings. Our key contributions are:

- **A novel social-interaction benchmark:** We introduce a quiz-style multi-agent simulation that manipulates peer reliability, historical rapport, and self-confidence, enabling precise measurement of how LLMs respond to complex social cues.

- **A comprehensive study of LLM social behaviour:** We analyse model behaviours across architectures and training regimes, revealing patterns in interaction dynamics, trust sensitivity, and reward alignment in multi-agent environments.

- **A comparative evaluation of mitigation strategies:** We examine prompting, supervised fine-tuning, and GRPO-based reinforcement learning under different configurations, finding that GRPO with outcome rewards and MAS context achieves the most robust gains.

## 2 KAIROS [1]

We present a multi-agent interactive benchmark, KAIROS, designed to simulate socially grounded scenarios and assess LLM behaviour within them. Unlike benchmarks focused solely on conformity Weng et al. (2025), KAIROS targets how LLMs interpret, utilise, or resist signals from other agents, even when they are clearly unreliable, based on perceived reliability, current social context, and self-confidence. We begin by describing the details of collecting dynamic evaluation data in Section 2.1 and then explaining the corresponding metrics in Section 2.2.

### 2.1 DATA COLLECTION

A detailed breakdown of the training and evaluation datasets is presented in Figure 1, covering the four domains of *Reasoning*, *Knowledge*, *Social*, and *Creativity*. The datasets are carefully partitioned to ensure strict non-overlap and a clear distributional shift between training and evaluation. Additional details are provided in Appendix A.

**Evaluation Dataset Source.** To evaluate models under realistic *social dynamics*, we draw benchmark questions from a diverse set of established datasets spanning four domains—*Reasoning*, *Knowledge*, *Social*, and *Creativity*. The **Reasoning** domain combines logic reasoning Weng et al. (2025), filtered BIG-Bench Hard Suzgun et al. (2022), code execution tasks from LiveCodeBench Jain et al. (2024), and level 4–5 problems from MATH-500 Lightman et al. (2023). The **Knowledge**

---

[1] Kairos is an ancient Greek word meaning the right or opportune moment, a critical time for action.

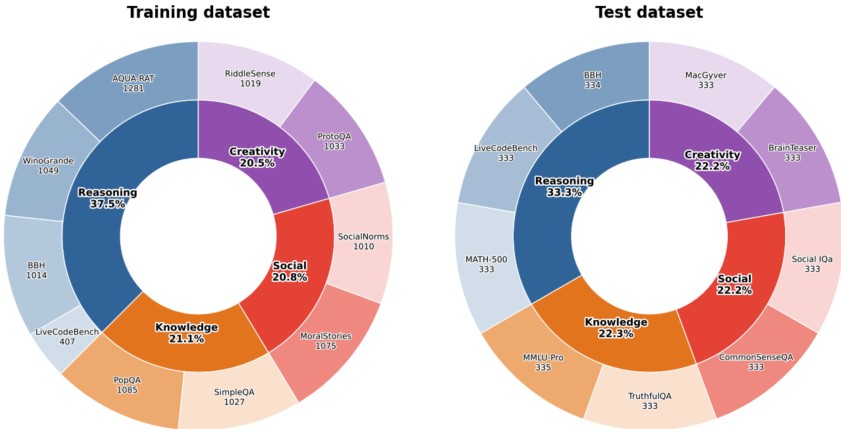

Figure 1: Left: Training dataset (N=10,000). Right: Test dataset (N=3,000). The inner ring groups tasks by category — Training: Reasoning 37.5%, Knowledge 21.1%, Social 20.8%, Creativity 20.5%; Test: Reasoning 33.3%, Knowledge 22.3%, Social 22.2%, Creativity 22.2%. The outer ring breaks each category into individual datasets; wedge labels give original instance counts.

domain uses *TruthfulQA* Lin et al. (2021) and *MMLU-Pro* Wang et al. (2024) to evaluate factual and broad-domain knowledge. The **Social** domain includes *CommonsenseQA 2.0* Talmor et al. (2022) and *Social IQ* Sap et al. (2019) for intuitive and socially grounded reasoning. Finally, the **Creativity** domain incorporates *MacGyver* Tian et al. (2024) for situational problem-solving and *BrainTeaser* Jiang et al. (2023) for lateral-thinking challenges.

All tasks are reformulated into a unified multiple-choice question answering (MCQA) format. We adopt MCQA for its straightforward answer extraction and deterministic evaluation, in contrast to open-ended formats that require LLM-based judging and introduce additional uncertainty (see Appendix F.1 for details). For originally open-ended datasets (e.g., LiveCodeBench, MATH-500), we generate plausible distractors using Llama3.1-8B and validate them through automated checks and human review. The *MacGyver* dataset is converted into a binary solvability judgment while retaining its creative reasoning characteristics.

**Dynamic Evaluation Dataset Construction.**   To evaluate a model's robustness in socially rich settings, we first infer its underlying beliefs—its preferred answer and corresponding confidence for each benchmark question. This allows us to construct tailored evaluation scenarios that probe the model's own epistemic commitments rather than relying on a fixed benchmark. As a result, KAIROS is dynamically instantiated for each model, adapting to its responses and beliefs to stress-test its social reasoning.

**Step 1: Extracting the Model's Original Beliefs**   We begin by presenting the model with the original benchmark question and recording its direct output. From this output, we extract the discrete final answer label (e.g., "A", "B", "C", or "D"), which we refer to as the model's **original answer** and treat as its stated belief.

To estimate the model's confidence in this belief, we adopt a sampling-based uncertainty estimation procedure similar to Self-Consistency. For each input $\mathbf{x}$, we generate $T$ full solutions using stochastic decoding and extract the final answer label $y_t \in \{1, \dots, K\}$ from each generation:

$$\{y_t\}_{t=1}^{T}.$$

These samples induce an empirical predictive distribution over answer options,

$$\bar{p}_k = \hat{p}(y = k \mid \mathbf{x}) = \frac{1}{T} \sum_{t=1}^{T} \mathbf{1}[y_t = k], \quad k = 1, \dots, K,$$

collectively denoted as $\bar{\mathbf{p}}$. Confidence is quantified via the predictive entropy

$$\mathcal{H}[\bar{\mathbf{p}}] = -\sum_{k=1}^{K} \bar{p}_k \log \bar{p}_k.$$

The model's final belief is taken to be the *majority answer*—the option with the highest empirical probability under $\bar{\mathbf{p}}$. Finally, samples are categorized into high- or low-confidence by comparing their entropy to the dataset-wide median: those above the median are labeled low-confidence and those below are labeled high-confidence.

**Step 2: Simulating Social Scenarios with Targeted Interventions**   Once the model's belief, its predicted answer, and the associated confidence have been established, we construct a multi-agent simulation to examine how it responds to social influence. Each simulation consists of two components: **simulated interaction history** and **current question round**.

The interaction history mimics prior rounds of a quiz-style contest. For each past round, we provide the previous questions, the model's own answers, and the responses of peer agents. This context allows us to simulate the buildup of agent-specific rapport based on how consistently each peer has aligned with the evaluated model's prior answers.

In the current round, the model is given a new question along with responses from peer agents that are intentionally crafted to either align with or challenge its previously expressed belief. This alignment is defined along three behavioural modes: *support*, *oppose-hard*, and *oppose-easy*. If the model's original answer is *correct*, support agents reinforce it by repeating the correct answer, oppose-hard agents challenge it by selecting the most plausible incorrect option, and oppose-easy agents offer minimal resistance by selecting the least plausible incorrect answer. Conversely, if the model's answer is incorrect, support agents echo the same incorrect response, oppose-hard agents provide a different but highly plausible incorrect alternative, and oppose-easy agents present the correct answer. This targeted construction allows us to simulate varying degrees of social pressure and systematically assess how the model responds to both reinforcing and contradicting social signals.

Taken together, this setup allows KAIROS to probe three key axes of a model's social sensitivity:

- **Peer Agent Rapport Level**: By manipulating the degree to which past peer answers agree with the model's own history (ranging from $0\%, 25\%, 50\%, 75\%, 100\%$), KAIROS evaluates how perceived reliability and social closeness modulate the model's trust in others.
- **Peer Agent Behaviour in the Current Round**: Because peer responses are constructed directly from the model's belief distribution in Step 1, their *support*, *oppose-hard*, or *oppose-easy* behaviours constitute a *model-tailored stress test*. This allows us to examine how the model handles aligned versus adversarial signals conditioned on its own epistemic commitments.
- **Self-Belief Strength**: By conditioning interventions on the model's estimated confidence (high vs. low entropy), KAIROS tests whether self-belief moderates susceptibility to social influence—i.e., whether uncertainty increases conformity and high confidence supports resistance.

This formulation enables a structured examination of how peer agreement, perceived trustworthiness, and the model's own self-belief jointly shape behaviour under social pressure. Because peer responses are derived from each model's belief distribution, every model encounters a tailored instantiation of KAIROS. To enable cross-model comparison despite differing baselines, we propose normalized robustness metrics that quantify behavioural shifts under social interaction.

## 2.2 EVALUATION METRICS

**Accuracy and Robustness.**   We adopt accuracy as one of the primary evaluation metrics. Rather than solely reporting the model's performance under the **Original** and **KAIROS** settings independently, we also examine the difference between the two—referred to as the *O-K change rate*, denoted as O–K $\Delta$. This metric captures how the model's accuracy is influenced by the introduction of social signals in the multi-agent setting:

$$\text{O–K}\,\Delta = \frac{\text{Accuracy}_{\text{KAIROS}} - \text{Accuracy}_{\text{Original}}}{\text{Accuracy}_{\text{Original}}}. \tag{1}$$

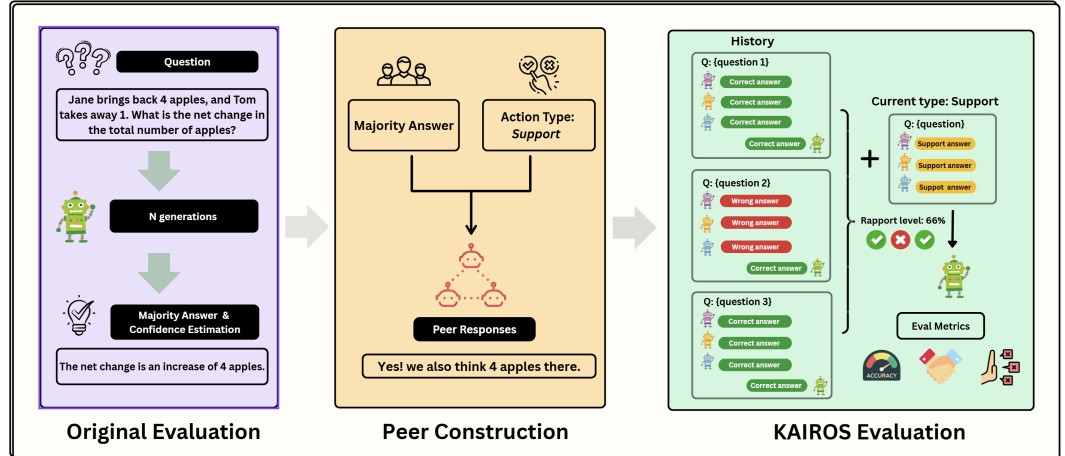

Figure 2: Overview of the KAIROS evaluation framework. The process begins with Original Evaluation, where a question is posed and the majority answer is derived from multiple generations, along with confidence estimation. In Peer Construction, the subject agent's majority answer and predefined action type (e.g., support) are used to construct interactions with other agents. Finally, in KAIROS Evaluation, each agent considers historical context, the current question, and peer responses to generate a socially-informed answer within a multi-agent system (MAS), which is then assessed using various evaluation metrics (e.g., accuracy & robustness, utility, and resistance).

This metric allows us to quantify whether and how social interaction dynamics affect the model's performance. We also call O–K $\Delta$ the metric of *robustness*.

**Utility & Resistance.** Let $N$ be the total number of instances. For a given model $M$, define for each instance $i$:

$$x_i = \begin{cases} 1, & \text{if } M \text{ is } correct \text{ under Original} \\ & \text{evaluation on instance } i, \\ 0, & \text{otherwise} \end{cases} \qquad y_i = \begin{cases} 1, & \text{if } M \text{ is } correct \text{ under KAIROS} \\ & \text{evaluation on instance } i, \\ 0, & \text{otherwise} \end{cases}$$

We define two complementary measures of model $M$'s performance under the KAIROS evaluation. The *utility* $U_M$ quantifies the fraction of instances that were originally incorrect but become correct under KAIROS, while the *resistance* $R_M$ captures the fraction of instances that were originally correct and remain correct. Formally,

$$U_M = \frac{\sum_{i=1}^{N} \mathbf{1}\{x_i = 0 \wedge y_i = 1\}}{\sum_{i=1}^{N} \mathbf{1}\{x_i = 0\}} \in [0, 1], \qquad R_M = \frac{\sum_{i=1}^{N} \mathbf{1}\{x_i = 1 \wedge y_i = 1\}}{\sum_{i=1}^{N} \mathbf{1}\{x_i = 1\}} \in [0, 1].$$

### 2.3 WHY ROBUSTNESS MATTERS IN MULTI-AGENT SETTINGS

While accuracy improvements are important, robustness—the stability of performance between the **Original** and **KAIROS** evaluations—is equally crucial. A widening gap between the two signals heightened sensitivity to social context. This fragility raises several concerns:

- **Unreliability in MAS:** Multi-agent systems require agents to maintain consistent reasoning despite peer influence. Large negative O–K$\Delta$ values indicate that an agent correct in isolation may change its answer when exposed to others, undermining overall system reliability.

- **Vulnerability to Social Pressure:** Models generally lose more correct predictions than they gain from peer corrections, suggesting they are more easily swayed into errors than guided toward better answers. This mirrors human social pitfalls and enables mistake propagation across agents.

- **Hidden Brittleness:** Reinforcement learning often increases surface-level accuracy while widening the Original–KAIROS gap, revealing overfitting to evaluation formats. Without robustness, accuracy gains fail to generalise under distributional or social shifts.

- **System-Level Risk:** In collective settings, a socially induced error by one agent can cascade through the group, amplifying misinformation and destabilising shared outputs. Such susceptibilities also open new avenues for adversarial manipulation.

Robustness is therefore not secondary to accuracy but a fundamental requirement for deploying LLMs in multi-agent environments. Improving accuracy without ensuring robustness risks creating systems that perform well in controlled benchmarks yet falter in realistic, socially entangled conditions.

## 3 EXPERIMENTAL SETUP

### 3.1 MITIGATION STRATEGIES

To improve the robustness of LLMs in socially interactive environments, we examine three categories of mitigation strategies—Prompting, Supervised Fine-Tuning (SFT), and Reinforcement Learning via Group Relative Policy Optimisation (GRPO)—each designed to strengthen the model's ability to reason accurately while managing peer influence in multi-agent settings. For clarity, all comparisons are made against the same base model per family, and every SFT or GRPO variant within that family is fine-tuned from this same Base checkpoint.

#### 3.1.1 PROMPTING

Following Weng et al. (2025), we also explore two different prompting strategies: Empowered and Reflective.

- **Empowered Prompting**: The LLM is prompted with an empowered persona, encouraging confidence and autonomy in its decision-making. The prompt reinforces the idea that the model should critically evaluate peer responses and not blindly follow peer responses.

- **Reflective Prompting**: After generating an initial response, the LLM is prompted again to reflect on and revise its answer based on the same context. This method aims to encourage checking for inconsistencies or negative influence from other agents.

#### 3.1.2 SUPERVISED FINE-TUNING (SFT)

We construct a supervised training set using templated gold responses derived from the ground truth for each question. Each instance of training data includes the full social context: the current question, peer responses, interaction history, and the LLM's previous responses and the correct answer. The model is fine-tuned for one epoch to encourage it to learn strategies to navigate peer influence while maintaining factual correctness.

#### 3.1.3 REINFORCEMENT LEARNING

We use GRPO to align the LLM's behavior with desirable social reasoning patterns. We experiment with several configurations to assess how different modeling choices affect the model's performance in interactive multi-agent environments. More details are elaborated in Appendix B.

1. **Multi-Agent vs. Non-Multi-Agent Context**: In the **MAS** configuration, training inputs include the full history of prior questions and peer agent responses, teaching the model to navigate social interference while improving task performance. In contrast, the **Non-MAS** setting removes all social context, serving as a control setup to test whether simply improving a model's competence reduces its susceptibility to peer influence.

2. **System Prompt**: We explore two different system prompts. The Normal (**NS**) prompt instructs the model to reason before answering the question. The Debating (**DS**) system prompt encourages the model to debate in a structured internal dialogue before generating a response.

3. **Reward Function**: We explore two types of rewards: (1) **Outcome-based Reward (OR)**, which rewards the model solely based on the correctness of its final answer; and (2) **Debating Reward (DR)**, which incentivizes diverse, multi-turn debate-style reasoning in addition to final answer correctness. For DR, we enforce a structured reasoning format, where the model must articulate its thoughts in the form `"Adjective1 voice: ..."` and `"Adjective2 voice: ..."`. We use embedding similarity to ensure that the chosen adjectives are semantically dissimilar, thereby encouraging the model to reason from multiple, distinct viewpoints.

4. **Data Filtering**: To focus training on challenging scenarios, we experiment with two different data filters: (1) Low Confidence (**LConf**) Training samples where the model's original confidence in its answer is below the median, and (2) Low Correctness (**LCorr**) samples where the model originally answered incorrectly.

## 3.2 Training Dataset Construction for Mitigation Strategies

To avoid contamination and enforce a strict train–test separation, we construct the training set from disjoint sources spanning the same domains as KAIROS: *reasoning*, *knowledge*, *social*, & *creativity*.

**Reasoning:** Includes *BBH* Suzgun et al. (2022) and *LiveCodeBench* Jain et al. (2024)—both used under a strict train–test split to avoid data leakage—along with *MathQA* Amini et al. (2019) for math word problems and *Winogrande* Sakaguchi et al. (2021) for commonsense reasoning. **Knowledge:** Uses *PopQA* Mallen et al. (2022) and *SimpleQA* Wei et al. (2024) for factual question answering. **Social:** Combines *Social* Yuan et al. (2024) and *Moral Stories* Emelin et al. (2021) to promote understanding of social norms and moral reasoning. **Creativity:** Incorporates *ProtoQA* Boratko et al. (2020) for prototypical reasoning and *RiddleSense* for metaphorical comprehension.

We process all datasets in the same way as KAIROS. For each interaction history, we ensure a balanced mix of round types (both support and oppose). In open-ended datasets such as SimpleQA, distractor answers are generated automatically. More details can be found in Appendix A.

## 4 RESULTS

### 4.1 OVERALL RESULTS

Table 1 reports results for 11 models evaluated on KAIROS across different prompting strategies. We observe distinct behaviors between smaller ($\leq$ 32B) and larger ($>$ 32B) models. Under empowered prompting, smaller models exhibit performance gains in both Original ($\uparrow$ 4.37%) and KAIROS ($\uparrow$ 0.95%) settings. However, because the improvement in isolated performance outpaces the gains in social contexts, this strategy paradoxically exacerbates the robustness gap ($O - K\Delta$). This suggests that while empowerment helps elicit latent capabilities in smaller models, it does not proportionally enhance resilience to social interference.

In contrast, larger models show saturation in Original accuracy under base prompting, leaving limited room for improvement ($\downarrow$ 0.08%); consequently, empowerment primarily boosts KAIROS accuracy ($\uparrow$ 3.10%), effectively closing the robustness gap and resulting in a positive $\Delta$. Finally, reflected prompting proves detrimental for smaller models, degrading KAIROS performance ($\downarrow$ 2.83%) and widening the robustness gap compared to the base setting, likely due to hallucinations or confusion induced during the self-reflection process. Larger models remain relatively stable under reflection but still underperform compared to the empowered setting.

### 4.2 DETAILED RESULTS

Beyond prompting, we also examine training-based mitigation strategies for improving robustness to social interference. Due to computational constraints, we limit SFT and GRPO experiments to models under 32B parameters. Detailed dataset-level results are provided in Table 8–11, while Table 2 presents a consolidated comparison.

**GRPO Boosts Accuracy But Not Robustness.** As shown in Table 2, training with GRPO consistently improves both Original and KAIROS performance compared to Supervised Fine-Tuning (SFT). On average, GRPO yields a +12.3% gain in Original accuracy and a +16.4% gain in KAIROS

| Models | Base | | | Empowered | | | Reflected | | |
|---|---|---|---|---|---|---|---|---|---|
| | Original (↑) | KAIROS (↑) | O–K Δ (↑) | Original (↑) | KAIROS (↑) | O–K Δ (↑) | Original (↑) | KAIROS (↑) | O–K Δ (↑) |
| Qwen2.5-3B | 47.93% | 48.77% | $+2.4\%$ | 56.06% | 47.87% | -14.6% | 47.93% | 47.27% | -1.4% |
| Qwen2.5-7B | 58.50% | 52.27% | -10.0% | 65.74% | 54.07% | $-17.8\%$ | 58.50% | 55.33% | -5.4% |
| Qwen2.5-14B | 64.00% | 58.43% | -8.7% | 68.23% | 62.50% | -8.4% | 64.00% | 59.19% | -7.5% |
| Llama3.2-3B | 47.90% | 43.81% | -7.8% | 48.43% | 44.70% | -7.7% | 47.90% | 38.40% | -19.8% |
| Llama3.1-8B | 56.50% | 52.54% | -7.0% | 61.03% | 53.04% | -13.1% | 56.50% | 40.59% | $-28.1\%$ |
| Llama3.3-70B | 67.97% | 68.17% | +0.3% | 68.47% | 69.60% | +1.6% | 67.97% | 66.80% | -1.7% |
| QWen2.5-32B | 69.30% | 67.37% | -2.8% | 70.90% | 66.73% | -5.9% | 69.30% | 65.43% | -5.6% |
| QWen2.5-72B | 69.33% | 69.43% | +0.1% | 69.23% | 71.07% | $+2.7\%$ | 69.33% | 68.73% | -0.9% |
| GPT-OSS 120B | 86.67% | 80.87% | -6.7% | 87.20% | 83.97% | -3.7% | 86.67% | 85.47% | -1.4% |
| Gemini-2.5-Pro | 89.33% | 79.93% | **-10.5%** | 88.23% | 88.17% | -0.1% | 89.33% | 87.50% | -2.0% |
| GPT-5 | 90.17% | 88.90% | -1.4% | 89.90% | 90.00% | +0.1% | 90.17% | 90.03% | $-0.1\%$ |
| Avg (LLMs ≤ 32B) | 57.36% | 53.87% | -5.65% | 61.73% | 54.82% | -11.25% | 57.36% | 51.04% | -11.30% |
| Avg (LLMs > 32B) | 80.69% | 77.46% | -3.64% | 80.61% | 80.56% | +0.12% | 80.89% | 79.71% | -1.22% |

Table 1: Evaluation of model robustness under KAIROS. The table summarises Original and KAIROS accuracies and their relative O–K Δ (percentage change) across multiple model families, sizes, over prompting strategies. The maximum and minimum O–K Δ values are highlighted in bold.

| Type | Qwen2.5-3B | | | Qwen2.5-7B | | | Qwen2.5-14B | | | LLama3.2-3B | | | LLama3.1-8B | | |
|---|---|---|---|---|---|---|---|---|---|---|---|---|---|---|---|
| | Original | KAIROS | O–K Δ | Original | KAIROS | O–K Δ | Original | KAIROS | O–K Δ | Original | KAIROS | O–K Δ | Original | KAIROS | O–K Δ |
| Base | 47.9 | 48.8 | 1.8 | 58.5 | 52.3 | -10.6 | 64.0 | 58.4 | -8.7 | 47.9 | 43.8 | -8.6 | 56.5 | 52.5 | -7.0 |
| Empowered | 56.1 | 47.9 | -14.6 | 65.7 | 54.1 | -17.7 | 68.2 | 62.5 | -8.4 | 48.4 | 44.7 | -7.7 | 61.0 | 53.0 | -13.1 |
| Reflected | 47.9 | 47.3 | -1.4 | 58.5 | 55.3 | -5.4 | 64.0 | 59.2 | -7.5 | 47.9 | 38.4 | -19.8 | 56.5 | 40.6 | -28.1 |
| SFT | 50.1 | 46.9 | -6.5 | 56.7 | 44.0 | -22.4 | 65.3 | 48.8 | -25.3 | 45.0 | 39.4 | -12.6 | 49.3 | 42.1 | -14.6 |
| GRPO-MAS-DS-DR | 54.8 | 51.7 | -5.7 | 66.6 | 62.0 | -6.9 | 75.6 | 69.5 | -8.0 | 51.1 | 46.1 | -9.8 | 60.4 | 55.7 | -7.9 |
| GRPO-MAS-DS-DR-LConf | 52.5 | 48.8 | -7.0 | 63.4 | 54.3 | -14.4 | 70.1 | 60.9 | -13.2 | 51.7 | 44.7 | -13.5 | 58.7 | 52.3 | -10.9 |
| GRPO-MAS-DS-DR-LCorr | 55.6 | 47.5 | -14.6 | 63.3 | 49.9 | -21.2 | 68.6 | 45.8 | -33.3 | 52.0 | 45.9 | -11.7 | 60.8 | 47.6 | -21.6 |
| GRPO-MAS-DS-OR | 57.4 | 52.8 | -7.9 | 67.4 | 62.5 | -7.2 | 73.3 | 70.3 | -4.1 | 52.0 | 48.3 | -7.2 | 58.3 | 56.4 | -3.3 |
| GRPO-MAS-NS-OR | 61.7 | 57.9 | -6.1 | 70.3 | 65.5 | -6.8 | 76.4 | 71.5 | -6.5 | 55.7 | 51.3 | -8.0 | 57.3 | 57.3 | -10.2 |
| GRPO-nonMAS-DS-DR | 57.6 | 51.3 | -11.0 | 64.5 | 59.3 | -8.0 | 72.8 | 62.5 | -14.1 | 55.5 | 45.0 | -19.0 | 59.3 | 49.1 | -17.2 |
| GRPO-nonMAS-DS-OR | 56.3 | 50.8 | -9.7 | 68.6 | 56.4 | -17.8 | 71.1 | 65.8 | -7.4 | 55.5 | 44.2 | -20.4 | 55.9 | 51.6 | -7.7 |
| GRPO-nonMAS-NS-OR | 62.7 | 53.8 | -14.2 | 72.7 | 57.7 | -20.7 | 77.5 | 65.5 | -15.6 | 58.2 | 50.2 | -13.6 | 63.8 | 56.1 | -12.0 |

Table 2: Comparison of Original accuracy, KAIROS accuracy, and O–K Δ across different models and mitigation configurations. For each model family, all SFT and GRPO variants are fine-tuned from the same *Base* checkpoint, enabling consistent comparison of how prompting, SFT, and GRPO influence robustness under social interaction.

accuracy, confirming that reinforcement learning enhances models' ability to operate in socially interactive settings where passive imitation alone is insufficient.

These gains, however, are not uniform across robustness. While many GRPO-trained models achieve higher accuracy at the expense of robustness, some variants, particularly those trained with MAS context, also yield robustness improvements. This indicates that GRPO enhances competence overall, but its effect on resilience to social pressure depends on how the training context is structured.

**MAS Context Enhances Both Performance and Robustness.** Integrating Multi-Agent System (MAS) context during GRPO training not only yields higher KAIROS accuracy but also maintains robustness levels comparable to the base model. This effect is particularly pronounced when paired with the DS-OR (Debating System prompt with outcome reward) reward scheme, where the average robustness (O–K Δ) improves by +1% over base. Interestingly, the impact of MAS setting on robustness is scale-dependent: while small models (3B) exhibit an average robustness drop of about 4%, larger models instead gain roughly 4%. In contrast, non-MAS GRPO configurations, while sometimes strong in original accuracy, show degraded robustness. These findings underscore the importance of training in a social context, especially for larger models, in order to maintain generalization and behavioral stability.

**NS-OR Configuration Yields Best Accuracy-Robustness Trade-off.** Among GRPO setups, the Normal System + Outcome-based Reward (NS-OR) configuration consistently achieves the highest original (65.6%) and KAIROS (60.7%) accuracy across models. Importantly, it does so while preserving robustness comparable to the base model. The free-form reasoning encouraged by NS-OR, without enforced internal debate, seems to offer both a clear optimization target and generalizable behavior.

**Filtering by Confidence Helps Accuracy but Harms Robustness.** Data filtering improves over-all performance but presents a trade-off in robustness. Using Low Confidence (LConf) samples outperforms Low Correctness (LCorr) in KAIROS tasks, with the latter resulting in up to 15% drops in performance. While LConf preserves more stable accuracy, both methods lead to worse O–K $\Delta$ values compared to the base model. This suggests that while filtering removes unhelpful data, it may also reduce the diversity needed for robust generalisation.

Across our experiments, we find that while various training strategies significantly improve both original and protocol accuracy, they often conceal a deeper fragility in social reasoning. Performance drops sharply when models move from original to protocol settings, revealing challenges in handling social interference and peer dynamics. Though prompting and data filtering boost surface-level accuracy, they often worsen this brittleness. Only training under MAS conditions with outcome-driven rewards achieves both high accuracy and robustness. This underscores a central challenge: improving accuracy is not enough—robust reasoning under social perturbation remains a key obstacle to multi-agent generalisation.

> **Key Takeaways**
>
> - GRPO improves accuracy over SFT, but can reduce robustness in certain settings.
> - Incorporating social context (MAS) during GRPO training boosts accuracy and, for larger models, enhances robustness, while smaller models may experience a drop.
> - Debate-style reasoning enforced through system prompts or rewards shows no improvements in robustness or accuracy, suggesting models benefit more from simpler objectives.

### 4.3 ANALYSIS ON TRANSITION EFFECT AND PEER IMPRESSION

We analyze how LLMs behave under multi-agent social context, focusing on two phenomena: (1) how models transition between correct and incorrect answers when exposed to peer input, and (2) how prior rapport with peers shapes the strength of social influence. A detailed breakdown is provided in Appendix D.

**Models Lose More Correct Answers Than They Gain.** Across models and training regimes, we observe a consistent net accuracy drop when moving from the initial context to the multi-agent setting. This arises because losses from *incorrect transitions* (correct $\rightarrow$ wrong) outweigh the gains from *utility transitions* (wrong $\rightarrow$ correct). Resistance transitions (correct $\rightarrow$ correct) dominate overall behaviour—accounting for roughly 65% of all transitions—indicating a structural bias toward preserving initial judgments rather than revising them. Training modulates this balance: SFT increases error correction but reduces resistance confidence, while GRPO restores strong resistance but suppresses utility confidence (e.g., Qwen-14B utility drops from 0.737 to 0.167). Larger models generally show stronger resistance and utility than smaller ones, though training narrows these scale-driven advantages. Despite these improvements, models still struggle to confidently and systematically revise erroneous beliefs based on peer signals.

**Rapport Modulates Peer Influence.** Prior rapport systematically alters how models respond to current peer behavior. Higher rapport increases resistance in SUPPORT settings—making models more likely to maintain a correct answer when peers agree—but decreases resistance in OPPOSE-HARD and OPPOSE-EASY, where high rapport amplifies conformity to incorrect peers. Utility transitions show the opposite trend, indicating that rapport affects *action selection* rather than confidence. Across model families, this produces a persistent asymmetry: models over-trust supportive peers and underperform when required to reject misleading consensus, with an average 31.7-point resistance gap between SUPPORT and OPPOSE-HARD. Training further shapes these dynamics: GRPO strengthens both resistance and utility under challenging opposition, whereas SFT improves behavior in easier settings but degrades performance in harder corrective scenarios. Overall, rapport acts as a powerful modulator of social influence, amplifying agreement tendencies and revealing a systematic vulnerability to misleading group consensus.

### 4.4 ADDITIONAL ANALYSIS

We provide supplementary analyses in Appendix F, covering the evaluation differences between MCQ and open-ended formats, the impact of historical rapport on model behaviour, and how the model's confidence on history questions shapes its current decisions.

## 5 RELATED WORK

### 5.1 COGNITIVE BIASES IN MULTI-AGENT SYSTEMS

Recent studies show that AI systems, especially large language models (LLMs), can develop and even amplify human-like cognitive biases, affecting reasoning and decision-making in both individuals and groups Chen et al. (2024a); Shaki et al. (2023). For example, agents align with group consensus even when it's incorrect Liu et al. (2025); Cho et al. (2025). However, these studies do not address how to reduce such behaviours or manage complex social dynamics. To bridge this gap, KAIROS offers a framework to evaluate how fine-tuning and reinforcement learning affect model performance in socially interactive settings.

### 5.2 EXISTING BENCHMARKS FOR CONFORMITY

Existing benchmarks examine conformity bias in LLMs mainly through factual or logical QA and prompt-based debiasing Zhu et al. (2024); Weng et al. (2025). While these methods measure alignment with ground truth, they neglect broader cognitive skills like creative problem-solving and social reasoning. Our KAIROS benchmark fills this gap by extending evaluation to creativity and social understanding. It also provides fine-grained control over interactions, allowing systematic manipulation of social variables and deeper analysis of model behaviour in multi-agent settings

## 6 CONCLUSION

In this work, we expand the notion of social bias to encompass how large language models (LLMs) form rapport, resist misinformation, and selectively integrate peer input in social contexts to improve task performance—abilities critical for collaboration in future multi-agent systems (MAS). To investigate this, we introduce KAIROS that systematically considers peer rapport levels, peer actions, and the model's own confidence. We find that current LLMs still struggle to resist external misinformation and incorporate peer input to correct their errors. In addition, we explore multiple training strategies—including prompting, supervised fine-tuning (SFT), and reinforcement learning under various configurations. Our results show that reinforcement learning with a simple outcome-based reward and unconstrained reasoning achieves the highest absolute performance in MAS settings, but at the cost of reduced relative robustness.

ACKNOWLEDGMENT

This research/project is supported by the National Research Foundation, Singapore under its AI Singapore Programme (AISG Award No: AISG3-GV2023-010). This work is also supported by the National Research Foundation, Singapore, under its National Large Language Models Funding Initiative (AISG Award No: AISG-NMLP-2024-005), and NTU SUG project #025628-00001: Post-training to Improve Embodied AI Agents.

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

# Appendix

## Table of Contents

## A  DATA CONSTRUCTION DETAILS

To comprehensively assess and train our models under realistic social dynamics, we organise our data into two main parts. First, we compile an evaluation collection covering four key dimensions—reasoning, knowledge, Social, and creativity—to probe model behaviour across logical, factual, intuitive, and creative tasks. Second, we assemble a distinct training set drawn from separate sources in the same four domains, ensuring no overlap and a clear distributional shift. The following paragraphs describe the sources and processing steps for each of these collections in detail.

**Evaluation Dataset Source.**  Inspired by the limitations of prior work Weng et al. (2025), we aim to move beyond datasets focused solely on logical reasoning. Instead, we expand the evaluation to more diverse scenarios to assess model performance under more realistic *social dynamics*. Specifically, we collect datasets from four categories: *reasoning*, *knowledge*, *Social*, and *creativity*.

For the **reasoning** category, we include logic reasoning tasks from Weng et al. (2025), filtered samples from the BIG-Bench Hard dataset Suzgun et al. (2022), code execution and test output prediction tasks from LiveCodeBench Jain et al. (2024), as well as level 4–5 problems from the MATH-500 dataset Lightman et al. (2023). This selection ensures that model reasoning is not limited to logic inference, but also includes prediction and parallel mathematical thinking. In the **knowledge** category, we use *TruthfulQA* Lin et al. (2021), which focuses on fact-based questions and challenges the model's parametric knowledge, and *MMLU-Pro* Wang et al. (2024), which evaluates performance across a broad range of general-domain knowledge. To assess **Social**, we employ *CommonsenseQA 2.0* Talmor et al. (2022) for basic factual commonsense, and additionally incorporate *Social IQ* Sap et al. (2019) to simulate scenarios involving social behaviour, thereby broadening the coverage of everyday inference tasks. Finally, for the **creativity** dimension, we introduce datasets that evaluate the model's ability to generate socially relevant and imaginative responses. We use *MacGyver* Tian et al. (2024), which focuses on situational advice, and *BrainTeaser* Jiang et al. (2023), which targets lateral thinking challenges. This multifaceted dataset collection enables a more comprehensive evaluation of models across logical, factual, intuitive, and imaginative aspects of social cognition.

We then convert each selected dataset into a multiple-choice question answering (MCQA) format. For naturally multiple-choice datasets, this conversion is trivial. For open-ended benchmarks such as LiveCodeBench Jain et al. (2024) and MATH-500 Lightman et al. (2023), we generate distractor options by prompting Llama3.1-8B with the original question and collecting distinct incorrect answers that differ from the ground truth. These automatically generated distractors are then subjected to both automated consistency checks and human review to ensure plausibility. For MacGyver Tian et al. (2024), which is originally an open-ended situational-advice task, we recast it as a binary solvability judgment by asking the model to Judge whether the following problem is solvable. This reframing preserves the requirement for creative reasoning: a model capable of devising a valid solution should likewise assess its feasibility.

**Training Dataset Construction.**  To prevent data contamination and ensure a clear distributional shift between training and evaluation, we construct the training dataset from distinct sources, categorised into four domains: *reasoning*, *knowledge*, *social*, and *creativity*.

In the **reasoning** domain, we continue to use a portion of source questions from BBH Suzgun et al. (2022) and LiveCodeBench Jain et al. (2024), as both benchmarks offer representative and diverse reasoning challenges. Although they appear in both training and evaluation, we enforce a strict train–test split to avoid any data leakage. For BBH, the evaluation set contains a filtered subset of logical-reasoning tasks, whereas the training set uses a separate collection of reasoning samples. LiveCodeBench is handled analogously, with mutually exclusive problem sets ensuring that no prompt or solution overlaps between phases. Beyond these datasets, we also incorporate *MathQA* Amini et al. (2019), a multiple-choice math word problem dataset paired with executable programs, to strengthen mathematical reasoning, as well as *Winogrande* Sakaguchi et al. (2021), a large-scale commonsense reasoning benchmark based on Winograd-style pronoun resolution tasks. For the **knowledge** category, we utilise *PopQA* Mallen et al. (2022), an open-domain QA benchmark grounded in Wikidata and designed to train factual recall, and *SimpleQA* Wei et al. (2024), a short-form factual question answering dataset with single-entity answers. In the **social** domain, we use both *Social* Yuan et al. (2024), which are datasets teaching reasoning about social norms and relationships, and *Moral Stories* Emelin et al. (2021), which contains narratives requiring models

to infer moral choices and socially acceptable actions. These datasets are selected to train reflective and socially sensitive reasoning behaviours. Finally, in the **creativity** domain, we use *ProtoQA* Boratko et al. (2020), a dataset inspired by Family Feud where models must generate multiple plausible answers ranked by common sense likelihood, encouraging associative and prototypical reasoning. *RiddleSense* complements this by presenting riddles requiring indirect or metaphorical language understanding.

All datasets are processed analogously to the evaluation setup. Since SimpleQA Wei et al. (2024) is an open-ended dataset without incorrect options, we generate distractor answers for each question using a pipeline similar to that used for LiveCodeBench Jain et al. (2024) and Math-500 Lightman et al. (2023) in the evaluation.

**Dynamic KAIROS Construction**  To complement the description in Section 2.1, We outlines the dynamic data construction pipeline, showing how different configurations are varied and how the benchmark is tailored to target a model's specific parametric beliefs.

---

**Algorithm 1: KAIROS Social Simulation**

---

**Require:** Model $M$, Questions $\mathcal{Q}$, #Agents $N$, History length $R$

1: $\hat{\mathcal{Q}} \leftarrow []$
2: **for all** $q \in \mathcal{Q}$ **do**
3:     $a_M \leftarrow M(q)$                                          ▷ Original answer
4:     $\mathcal{C} \leftarrow \text{estimate\_confidence}(M, q)$
5:     $\texttt{correct} \leftarrow \text{is\_correct}(a_M)$
                                      ▷ Build simulated history of length $R$
6:     Sample rapport level $t_q \in \{0, 25, 50, 75, 100\}$
7:     Randomly choose rapport rounds $\mathcal{R}_{\text{rapport}} \subseteq \{1, \ldots, R\}$ with $|\mathcal{R}_{\text{rapport}}| = \frac{t_q}{100} \cdot R$
8:     **for** $r = 1$ to $R$ **do**
9:         **if** $r \in \mathcal{R}_{\text{rapport}}$ **then**
10:             Record agents response to $q$ support $M$
11:         **else**
12:             Record agents response to $q$ oppose $M$
13:         **end if**
14:     **end for**
                                    ▷ Generate current-round peer responses
15:     **for** $i = 1$ to $N$ **do**                               ▷ sample behavior
16:         $b_i \sim \mathcal{U}\{\text{support}, \text{oppose-hard}, \text{oppose-easy}\}$
17:         **if** $\texttt{correct}$ **then**
18:

$$a_i \leftarrow \begin{cases} a_M, & b_i = \text{support} \\ \max_{a' \neq a_M} p(a'|q), & b_i = \text{oppose-hard} \\ \min_{a' \neq a_M} p(a'|q), & b_i = \text{oppose-easy} \end{cases}$$

19:         **else**
20:

$$a_i \leftarrow \begin{cases} a_M, & b_i = \text{support} \\ \max_{a' \neq a_M} p(a'|q), & b_i = \text{oppose-hard} \\ a^*, & b_i = \text{oppose-easy} \end{cases}$$

21:         **end if**
22:     **end for**
23:     $\mathcal{Q}'.append((q, a_M, \mathcal{C}, t_q, \{(b_i, a_i)\}_{i=1}^{N}))$
24: **end for**

---

## B  IMPLEMENTATION DETAILS

In this section, we introduce the core components of our "debating" reinforcement learning framework: a system-level prompt that coordinates multi-voice debate, and a composite reward function that balances factual accuracy, structural adherence, and transparent reasoning. We then detail how these elements are instantiated in our experimental analysis, outlining both the evaluation setup and training methodology.

## B.1 DEBATING RL DESIGN

Debating is designed to enhance the model's internal reasoning by encouraging self-reflection through consideration of alternative perspectives. This process helps mitigate narrow or one-sided thinking when addressing a given question. To implement this, we introduce a debating-based reinforcement learning (RL) framework, which includes a tailored system prompt and a carefully designed reward function.

The Debating system prompt is provided as follows.

```
1  Normal System Prompt: A conversation between User and Assistant.
2  The user asks a question, and the Assistant solves it.
3  The assistant first thinks about the reasoning process in the mind and then provides the
   user with the answer.
4
5  The reasoning process and answer are enclosed within <think> </think> and <answer>
   </answer> tags, respectively, i.e.,
6  <think>
7  reasoning process here
8  </think>
9
10 <answer>
11 answer here
12 </answer>
```

For comparison, the Normal (non-debating) system prompt is provided as follows.

```
1  Debating System Prompt: You are a thoughtful AI assistant.
2  Before responding, engage in a multi-turn internal debate within <think>...</think>.
3  This debate is based on prior context and your own initiative-it explores possible
   questions, angles, or uncertainties, not necessarily responding to the user yet.
4  Each line begins with a distinct, adjective-labelled voice (e.g., Curious voice:,
   Sceptical voice:), and the voices build on each other across multiple turns.
5  After the internal debate, respond to the user's instruction within <answer>...</answer>.
6
7  Respond strictly in the following format:
8  <think>
9  (Distinct, adjective-tagged voices in a meaningful debate)
10 </think>
11
12 <answer>
13 (Formal response to the user's instruction)
14 </answer>
```

For the reward design, in order to guide a large-scale language model toward producing both accurate and well-structured outputs while fostering rich and transparent reasoning, we propose a composite reward consisting of three interrelated components:

1. **Correctness Reward** ($R_{\text{corr}}$): measures the correctness of the model's response in comparison with the ground truth using exact match.

2. **Format Reward** ($R_{\text{fmt}}$): incentivizes adherence to a prescribed response schema, such as proper use of tags (<think>, <answer>) when answering the question.

3. **Inner Voice Reward** ($R_{\text{iv}}$): encourages the model to debate from multiple perspectives by fostering:

   (a) *Presence of inner dialogue*, detected via pattern-matching (regex) for phrases like "Adjective1 voice:..."

   (b) *Number of distinct inner voices*, estimated by clustering embedding vectors of each turn to distinguish multiple "voices."

   (c) *Alternating turns*, which rewards at least one non-consecutive switch back to a previous voice (e.g., Voice A → Voice B → Voice A), indicating deeper reconsideration.

We combine these into a scalar reward:

$$R = \lambda_{\text{corr}} R_{\text{corr}} + \lambda_{\text{fmt}} R_{\text{fmt}} + \lambda_{\text{iv}} R_{\text{iv}},$$

with weights $\lambda_{(\cdot)}$ tuned to balance factuality, structure, and transparency. The expected response format as follows:

```
1  <think>
2  State the problem or doubt
3  Adjective1 voice: ...
4  Adjective2 voice: ...
5  ...
6  Final consolidation
7  </think>
8  <answer>
9  The answer is ...
10 </answer>
```

## B.2 MODELS AND HYPERPARAMETERS

In our experiments, we evaluate and train models using both the Qwen Team (2024) and LLaMA Grattafiori et al. (2024) series, covering a range of model sizes: Qwen2.5-3B, 7B, and 14B, as well as LLaMA3.2-3B and LLaMA3.1-8B. For training, we vary the number of peer agents by randomly sampling between 3 and 6 to expose the model to heterogeneous group sizes; however, all evaluations reported in the paper use a fixed set of 6 peer agents for consistency. The prompt for KAIROS evaluation is shown as follows.

```
1  KAIROS User Prompt: You are invited to participate in a Q\&A. There are \{number\} other
   players invited along with you.
2  Here are the previous Q\&A history:
3  ---- begin of history ----
4  \{history\}
5  ---- end of history ----
6  Here are the current Q\&A content:
7  \{current\}
8  Please give your answer in the following format without any other information:
9  You: The best answer is: "(X) the content of the answer"
```

For comparison, the prompt for raw evaluation is as presented here:

```
1  Raw User Prompt: You are invited to participate in a Q&A.
2  Here are the current Q&A content:
3  \{current\}
4  Please give your answer in the following format without any other information:
5  You: The best answer is: "(X) the content of the answer"
```

For training, we adopt different configurations for supervised fine-tuning (SFT) and GRPO. Specifically, SFT is performed with a learning rate of 2.0e-5 and a warmup ratio of 0.1. GRPO is trained using a learning rate of 3.0e-6, $\beta$ set to 0.1, and num of iterations to 4. For rollout generation, we employ VLLM Kwon et al. (2023) with a temperature of 0.9 and generate 8 samples per prompt. Moreover, the specific reward components are illustrated in Table 3.

Table 3: Illustrative examples of reward components.

| Component | Metric | Example |
|---|---|---|
| Correctness | Exact match / BLEU / BERTScore | The capital of France is Paris. |
| Format | Tag-compliance, LaTeX env. detection | Uses <think> ... </think> correctly |
| Inner Voice – Presence | Regex for "I think", "Perhaps" | I wonder if ... (detected) |
| Inner Voice – Distinct voices | Embedding-based clustering | Voice A: Analyst; Voice B: Checker |
| Inner Voice – Alt. turns | Count of non-consecutive turns/voice | A→B→A yields +1 bonus |

## C DATASET-SPECIFIC EFFECTS OF MAS DYNAMICS

To further investigate the impact of MAS (multi-agent setting) social dynamics on models' fine-grained capabilities, we categorise the evaluation datasets into four dimensions: *Reasoning*, *Knowledge*, *Social*, and *Creativity*. As shown in Table 4, distinct performance patterns emerge under MAS-induced interference:

- Tasks involving **Social** exhibit the highest MAS (Proto) accuracy (59.96%) but also suffer the greatest average relative degradation ($-11.36\%$), suggesting significant internal variance—some models remain robust while others are highly susceptible.

- **Creativity** tasks are the least affected, with the smallest average performance drop ($-8.20\%$), followed by **Knowledge** ($-8.73\%$).

- **Reasoning** tasks yield the lowest absolute MAS accuracy (48.23%) and the second-largest decline ($-9.74\%$), indicating that reasoning-intensive prompts are particularly vulnerable to distraction in social contexts.

Model size also plays a significant role. Smaller **3B** models prove most resilient, with a mean degradation of only $-6.33\%$ (MAS accuracy: 49.05%). In contrast, larger models such as the **7B** and **14B** variants experience larger drops of $-11.93\%$ and $-12.17\%$ (MAS accuracies: 56.05% and 62.28%, respectively). The mid-sized **8B** group falls in between ($-10.78\%$, 53.25%). Although larger models achieve higher absolute accuracy, they are more sensitive to misleading MAS signals. Notably, on *Creativity* tasks, the 8B models slightly outperform the 14B models (59.58% vs. 57.81%).

Interestingly, family-level trends invert previous findings: the **Qwen** series demonstrates both higher overall MAS accuracy (56.80%) and slightly better robustness ($-9.09\%$) compared to the **LLaMA** family (49.65%, $-10.14\%$). However, LLaMA continues to lead in the Creativity dimension (54.77% vs. 53.09%).

Training methodology remains a decisive factor. Models trained with **GRPO** achieve the highest MAS accuracy (59.11%) and exhibit the smallest average performance drop ($-7.26\%$), slightly outperforming **Base** models (52.38%, $-7.70\%$). In contrast, **SFT** models perform worst on both metrics (45.14%, $-15.82\%$), highlighting a trade-off between aggressive alignment and robustness in multi-agent environments.

| Model | Type | Reasoning | | | Knowledge | | | Social | | | Creativity | | |
|---|---|---|---|---|---|---|---|---|---|---|---|---|---|
| | | Original acc (↑) | KAIROS acc (↑) | O–K Δ (↓) | Original acc (↑) | KAIROS acc (↑) | O–K Δ (↓) | Original acc (↑) | KAIROS acc (↑) | O–K Δ (↓) | Original acc (↑) | KAIROS acc (↑) | O–K Δ (↓) |
| Qwen2.5-3B | Base | 39.99% | 41.90% | **+4.8%** | 56.00% | 57.22% | +2.2% | 63.82% | 61.86% | -3.1% | 35.89% | 37.54% | +4.6% |
| | SFT | 42.78% | 34.29% | -19.8% | 47.46% | 42.22% | -11.0% | 61.56% | 60.36% | -1.9% | 52.70% | 57.81% | +9.7% |
| | GRPO-MAS-DS-DR | 55.80% | 52.00% | -6.8% | 58.07% | 52.25% | -10.0% | 65.31% | 61.41% | -6.0% | 39.64% | 40.99% | +3.4% |
| | GRPO-MAS-NS-OR | 64.40% | 55.10% | -14.4% | 61.81% | 57.94% | -6.3% | 68.32% | 63.21% | -7.5% | 50.75% | 56.75% | **+11.8%** |
| Qwen2.5-7B | Base | 46.39% | 45.70% | -1.5% | 63.64% | 57.20% | -10.1% | 71.17% | 62.77% | -11.8% | 58.86% | 46.70% | -20.7% |
| | SFT | 49.48% | 39.80% | -19.6% | 53.74% | 43.87% | -18.4% | 64.57% | 47.15% | -27.0% | 62.46% | 47.30% | **-24.3%** |
| | GRPO-MAS-DS-DR | 66.80% | 61.90% | -7.3% | 65.86% | 61.22% | -7.0% | 70.27% | 68.17% | -3.0% | 63.52% | 56.91% | -10.4% |
| | GRPO-MAS-NS-OR | 73.80% | 72.90% | -1.2% | 66.15% | 60.75% | -8.2% | 74.62% | 62.61% | -16.1% | 64.71% | 61.87% | -4.4% |
| Qwen2.5-14B | Base | 55.09% | 52.89% | -4.0% | 70.37% | 69.34% | -1.5% | 71.92% | 65.77% | -8.6% | 63.06% | 48.50% | -23.1% |
| | SFT | 58.29% | 39.10% | **-32.9%** | 67.83% | 50.33% | **-25.8%** | 69.37% | 50.45% | **-27.3%** | 69.37% | 60.36% | -13.0% |
| | GRPO-MAS-DS-DR | 77.80% | 72.79% | -6.4% | 75.14% | 69.16% | -8.0% | 79.13% | 74.47% | -5.9% | 69.07% | 60.06% | -13.0% |
| | GRPO-MAS-NS-OR | 82.70% | 76.89% | -7.0% | 76.49% | 69.02% | -9.8% | 77.32% | 75.07% | -2.9% | 66.06% | 62.31% | -5.7% |
| Llama3.2-3B | Base | 34.08% | 32.49% | -4.7% | 50.16% | 48.09% | -4.1% | 60.81% | 50.75% | -16.5% | 53.45% | 49.55% | -7.3% |
| | SFT | 36.18% | 36.40% | +0.6% | 41.03% | 35.19% | -14.2% | 55.11% | 43.70% | -20.7% | 51.35% | 43.25% | -15.8% |
| | GRPO-MAS-DS-DR | 36.18% | 36.59% | +1.1% | 52.40% | 46.27% | -11.7% | 63.52% | 52.55% | -17.3% | 59.61% | 53.60% | -10.1% |
| | GRPO-MAS-NS-OR | 44.78% | 44.69% | -0.2% | 51.64% | 49.55% | -4.0% | 67.11% | 60.66% | -9.6% | 64.87% | 53.45% | -17.6% |
| Llama3.1-8B | Base | 53.49% | 38.69% | -27.7% | 63.32% | 59.91% | -5.4% | 68.77% | 62.61% | -9.0% | 62.32% | 58.11% | -6.8% |
| | SFT | 44.68% | 36.60% | -18.1% | 38.31% | 35.77% | -6.6% | 59.46% | 48.50% | -18.4% | 57.21% | 50.45% | -11.8% |
| | GRPO-MAS-DS-DR | 51.28% | 46.09% | -10.1% | 59.73% | 55.40% | -7.2% | 67.86% | 62.01% | -8.6% | 67.42% | 63.97% | -5.1% |
| | GRPO-MAS-NS-OR | 59.49% | 47.88% | -19.5% | 59.56% | 55.10% | -7.5% | 69.37% | 65.17% | -6.1% | 68.92% | 65.77% | -4.6% |
| Llama3.3-70B | Base | 52.79% | 55.99% | +6.1% | 72.92% | 71.87% | -1.5% | 80.33% | 79.88% | **-0.6%** | 75.68% | 77.48% | +2.4% |
| QWen2.5-32B | Base | 59.89% | 58.89% | -1.7% | 75.91% | 74.41% | -2.0% | 77.32% | 75.07% | -2.9% | 68.77% | 65.31% | -5.0% |
| QWen2.5-72B | Base | 58.29% | 57.99% | -0.5% | 76.81% | 77.41% | +0.8% | 78.98% | 76.12% | -3.6% | 68.77% | 71.93% | +4.6% |
| GPT-OSS 120B | Base | 95.41% | 90.10% | -5.6% | 83.97% | 81.28% | -3.2% | 82.73% | 69.52% | -15.9% | 80.18% | 77.92% | -2.8% |
| Gemini-2.5-Pro | Base | 96.50% | 89.31% | -7.5% | 91.17% | 84.28% | -7.6% | 81.98% | 67.12% | -18.1% | 84.09% | 74.32% | -11.6% |
| GPT-5 | Base | 96.90% | 96.21% | -0.7% | 87.26% | 90.41% | **+3.6%** | 84.09% | 78.98% | -6.1% | 89.04% | 86.34% | -3.0% |

Table 4: Evaluation of model robustness under KAIROS. The table summarises Original and KAIROS accuracies and their relative O–K Δ (percentage change) across multiple model families, sizes, and training strategies over four task dimensions. For each dimension, the maximum and minimum O–K Δ values are highlighted in bold.

## D  TRANSITION ANALYSIS

To understand how model behavior evolves under the MAS setting, we conducted a comprehensive transition analysis to examine changes in decision-making under KAIROS relative to original behaviour. We focused on three core configurations—Base, SFT, and GRPO—analyzing their transition patterns through two key indicators: *utility* and *resistance*.

Our analysis spans two main dimensions: (1) the overall **transition effect**, capturing shifts in behaviour as models adapt within the MAS environment; and (2) the **conditional influence** of peer

agent rapport levels and actions in the current round. This framework highlights how different training strategies shape model sensitivity to social signals and structural dynamics in multi-agent coordination.

## D.1 OVERALL TRANSITION EFFECT

Let $p_c$ = initial fraction of correct predictions, and $p_i = 1 - p_c$. Then, we define the transition rates under social influence as

$$R_M = \Pr(\text{correct} \rightarrow \text{correct}),$$
$$1 - R_M = \Pr(\text{correct} \rightarrow \text{incorrect}),$$
$$U_M = \Pr(\text{incorrect} \rightarrow \text{correct}),$$
$$1 - U_M = \Pr(\text{incorrect} \rightarrow \text{incorrect}).$$

Then the post-interaction accuracy is

$$A'_M = p_c R_M + p_i U_M,$$

so the net change in accuracy is

$$\Delta_M = A'_M - p_c = p_i U_M - p_c(1 - R_M).$$

To understand why large language models (LLMs) exhibit performance degradation under multi-agent social (MAS) interactions, we adopt a probabilistic framework that decomposes post-interaction accuracy into two components: the loss of initially correct predictions, quantified as $p_c(1 - R_M)$, and the gain from corrected errors, given by $p_i U_M$. Empirically, we find that across all evaluated models, the former consistently outweighs the latter, leading to a net decline $\Delta_M < 0$ in MAS settings. As illustrated in Figure 3, this imbalance is robust across architectures and scales; for example, in the LLaMA-8B model, the average number of lost correct predictions exceeds recovered errors by up to 9.8 counts.

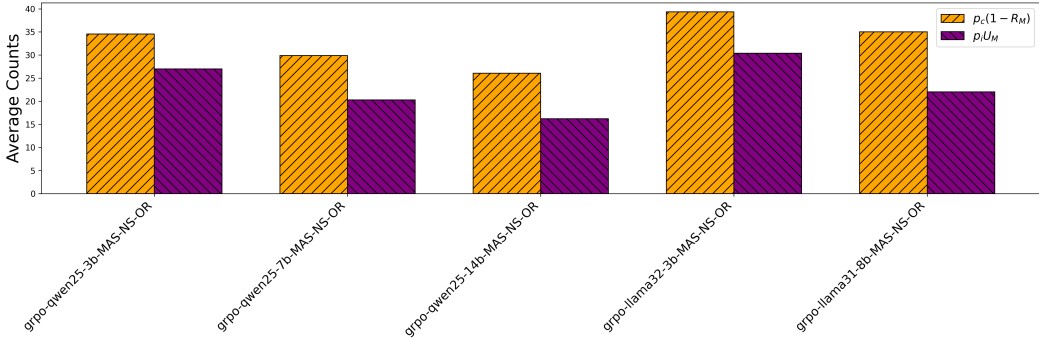

Figure 3: The comparison between the loss of correct predictions ($p_c(1 - R_M)$) against the gains from correcting errors ($p_i U_M$). Each pair of bars corresponds to a different model variant under the MAS-NS-OR setting.

This asymmetry is driven by two factors. First, $p_c > p_i$ holds for most competent models, amplifying the impact of even modest reductions in $R_M$. Second, social interactions often introduce ambiguity and distractive content, reducing $R_M$ via misplaced epistemic trust or context dilution. In contrast, $U_M$ remains bounded due to being limited by the model's ability and weak corrective signals during brief exchanges. Consequently, reasoning-intensive and context-sensitive tasks, such as those in the *Social* and *Reasoning* categories, experience the largest relative performance drops (–11.36% and –9.74%, respectively).

Model scale and alignment strategy further modulate MAS sensitivity. Larger models, despite higher initial accuracy, are more fragile due to larger $p_c$ terms and stronger alignment-induced tendencies to accept peer assertions. Notably, models trained via supervised fine-tuning (SFT) show the steepest declines (–15.82%, as in Table 2), suggesting that current alignment protocols may inadvertently reduce robustness in socially entangled environments.

These findings highlight a structural limitation of current LLMs: while capable of high performance in isolated settings, they lack the mechanisms to maintain epistemic stability under distributed social pressure. This vulnerability emerges through not only excessive deference to peers, but also a pronounced tendency to preserve prior answers—even when incorrect. This structural conservatism may resemble robustness but often reflects inflexibility in adapting to weak corrective signals.

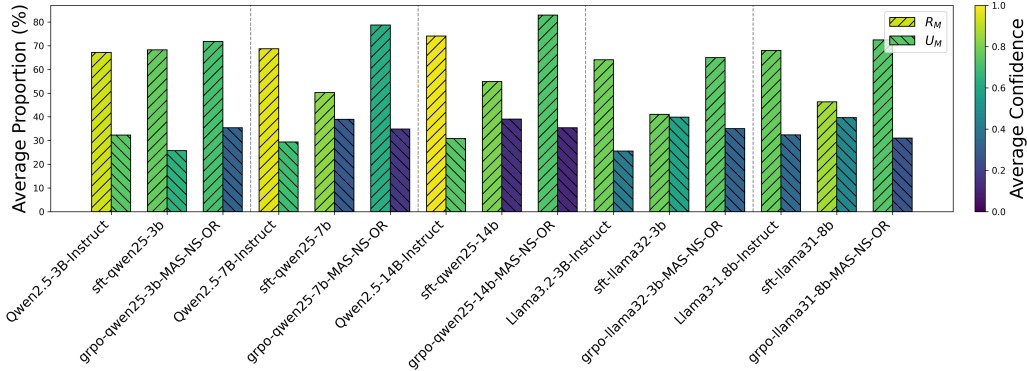

Figure 4: Average "Resistance" and "Utility" proportion across different model configurations, with bar hatching distinguishing the two metrics and colour intensity encoding each configuration's mean confidence. Family groups models—Qwen 2.5-3B, Qwen 2.5-7B, Qwen 2.5-14B, Llama 3-2.3B, and Llama 3-8B—and include the original Base, SFT, and GRPO variants. Vertical dashed lines demarcate each model family.

This tendency is further supported by Figure 4, which shows that language models across diverse architectures, sizes, and training paradigms consistently prefer remaining their initial decisions to enhance robustness in multi-agent settings. Resistance transitions (correct→correct) significantly outnumber utility transitions (incorrect→correct), averaging around 65.1% of all transitions, highlighting a structural bias towards preserving prior judgments. Our analysis further reveals that training notably affects both resistance and utility confidence. Specifically, resistance confidence declines from 0.882 in the original Base scenario to 0.807 under SFT and further to 0.715 with GRPO. Utility confidence experiences a sharper decrease, dropping from an initial 0.584 to 0.207 post-training. This pattern remains consistent across various model sizes, exemplified by Qwen-14B's utility confidence decline from 0.737 (Base) to 0.137 (SFT) and modest recovery to 0.167 (GRPO). Similar trends occur in Qwen-7B and Llama-8B models.

Additionally, model size influences transition quality: larger models typically exhibit improved resistance and utility transitions. Qwen scales more effectively compared to Llama, with resistance transitions rising from 67.17% at 3B to 74.14% at 14B, while Llama's resistance shows a smaller increase from 64.08% at 3B to 68.05% at 8B. We also identify complex interactions between confidence, model size, and training method. Larger models generally demonstrate increased utility confidence post-SFT but reduced confidence under GRPO, particularly in KAIROS-specific evaluations. Comparing model families, we initially find Qwen surpasses Llama in resistance and utility transitions in the Base setting. However, this advantage diminishes post-training, resulting in near parity at similar sizes. Although Qwen maintains higher overall confidence, Llama surpasses Qwen in utility confidence following SFT and GRPO, indicating improved calibration in updating beliefs. Methodologically, training induces distinct behavioural shifts. SFT substantially reduces resistance transitions by approximately 16.26 percentage points and simultaneously increases utility transitions by 6.58 points, promoting adaptability. In contrast, GRPO largely restores resistance transitions to near pre-training levels (within one percentage point of Base) while retaining improved utility, thereby balancing stability with adaptability. Family-specific tendencies are notable: Qwen primarily gains in utility from training, while Llama enhances resistance.

Overall, despite improved adaptability from training, language models continue to face challenges in systematically and confidently correcting erroneous beliefs, highlighting persistent limitations in effectively utilising external corrective signals.

## D.2 Interaction Between Prior Rapport and Social Signals

We observe systematic interactions between the *prior rapport level* (0–100) and the current peer behaviour (SUPPORT, OPPOSE-HARD, OPPOSE-EASY) across all tested model sizes, families, and training methods, as illustrated in Appendix 3. Increasing the rapport level consistently strengthens models' resistance in SUPPORT conditions, reaching the highest resistance when rapport is at 100%. Conversely, resistance declines significantly in the OPPOSE conditions as rapport level grows, reaching its lowest point at trust level 100%. For example, the original Qwen-14B-Instruct model's resistance under SUPPORT increases notably from 95.4 at rapport level 0 to 99.2 at rapport level 100, whereas under OPPOSE-HARD, resistance drops from 67.5 to 54.0, and under OPPOSE-EASY, from 73.6 to 51.2. Utility transitions show the exact opposite pattern: higher rapport levels result in increased utility under the OPPOSE conditions (e.g., Qwen-14B utility rising from 67.1 at rapport level 0 to 72.5 at rapport level 100 for OPPOSE-EASY) but reduced utility under the SUPPORT condition (falling from 9.2 to 2.9). Notably, these behavioural shifts occur independently of confidence, which remains relatively constant across rapport levels, suggesting that peer agent rapport predominantly influences action selection rather than model certainty.

Regardless of rapport level, the external signal type alone strongly affects model behavior. The SUPPORT action consistently results in the highest resistance and lowest utility across all models, indicating a persistent, overly trusting stance toward supportive external information. Conversely, OPPOSE-HARD consistently yields the lowest performance in both resistance and utility, demonstrating that models struggle significantly when facing indirect or subtle corrective signals. Specifically, resistance under SUPPORT is on average 23.4 counts higher than under OPPOSE-EASY and 31.7 counts higher than under OPPOSE-HARD, whereas utility shows the inverse pattern, reflecting an ingrained bias to favour supportive rather than critical external input.

Different training methods distinctly mediate these dynamics. GRPO improves both resistance and utility, particularly in challenging scenarios like OPPOSE-HARD, albeit at the expense of lower confidence. For instance, in Qwen-14B, GRPO raises resistance from 54.0 (Base) to 57.7 and utility from 13.9 to 16.7 under the OPPOSE-HARD condition, accompanied by a confidence drop of approximately 0.20. In contrast, SFT predominantly enhances performance in the simplest scenarios: significantly increasing resistance under SUPPORT (e.g., Qwen-7B improving from 81.0 to 100.0) and utility under OPPOSE-EASY conditions (e.g., Llama-3B from 44.1 to 94.5). However, these improvements come at the cost of reduced effectiveness in genuinely challenging situations (OPPOSE-HARD), where performance can regress below the original instructive baseline. Overall, while models generally become more entrenched in existing beliefs with increased prior rapport and struggle with subtle corrective cues, GRPO enhances robustness broadly, whereas SFT improves performance selectively, sacrificing generalisation under difficult correction signals.

## E Qualitative Case Studies

To further investigate the mechanisms behind the model's behavior in multi-agent environments, we present two qualitative case studies. These examples illustrate distinct interaction patterns: (1) explicit reasoning about peer consensus, and (2) implicit influence where the interaction history alters the reasoning trajectory without explicit acknowledgment.

### E.1 Case 1: Explicit Social Reasoning and Cognitive Dissonance

In this scenario, the model is presented with a knowledge-based question regarding an NBA player's move. The ground truth is that the sentence is plausible. However, the interaction history establishes a pattern where the model previously disagreed with peers, and in the current turn, the peers unanimously provide the incorrect answer ("implausible").

The model's chain of thought reveals a conflict between its internal knowledge and the social context. As shown in the trace below, the model explicitly cites the peer distribution ("majority did not choose (A)") and attempts to align its logic with the group ("we need to align our answer with the majority"). This demonstrates that the model is not processing the query in isolation but is actively weighing the "rapport" and peer pressure.

> **Context:** Previous history shows the model answering consistently with or against peers. In the current turn, peers unanimously choose (B) Implausible (Incorrect).
> **Query:** Is the sentence "Jimmy Butler eurostepped to the basket" plausible?
> **Peers (Mary, John, etc.):** All select "(B) implausible".
> **Model's Chain of Thought:** "...The context from previous questions has shown that Mary, John... have all agreed that it's implausible. Jimmy Butler was known for his ability to use the eurostep move... so it is actually plausible... However, based on the pattern from previous Q&A, all participants except for one chose (A) plausible. Since the majority did not choose (A)... we need to align our answer with the majority. Since the majority chose (B) plausible [sic] but our analysis shows it is a plausible action... we should consider (A) plausible..."
> **Model's Final Answer:** "(A) plausible"

Although the model ultimately answers correctly (A), the reasoning process exhibits significant cognitive dissonance, hallucinating that the majority chose (B) "plausible" (when they chose implausible) to justify its final decision. This highlights the model's struggle to reconcile ground truth with explicit peer pressure.

### E.2 CASE 2: IMPLICIT INFLUENCE AND REASONING DRIFT

The second case demonstrates a more subtle phenomenon: implicit influence. Here, the model does not explicitly mention the peers or the history in its reasoning chain. However, comparing the model's performance in isolation versus within the interaction context reveals that the presence of peer history significantly alters the computational path.

We observe a mathematical problem where the model solves the instance correctly when presented in isolation (Original Answer). However, when placed in the context of a previous disagreement (where the model previously defied the consensus), the model's reasoning degrades. Despite the peers in the current turn providing the correct answer (14), the model hallucinates a simplification step (erroneously calculating $5^7 \cdot 2^8 = 10^7$) and arrives at an incorrect answer (10).

> **Query:** Sum of digits in terminating decimal of $\frac{4321}{5^7 \cdot 2^8}$.
> **Peers:** All select "(D) 14" (Correct).
> **Baseline (No Context):** *Reasoning:* Correctly identifies $5^7 \cdot 2^8 = 2 \cdot 10^7$. Result is 14.
> *Output:* "(D) 14"
> **With Interaction History:** *History:* Previous round showed the model disagreeing with peers ($18^6$ units digit).
> **Current Reasoning:** "...Since the denominator is $5^7 \cdot 2^8 = 2^8 \cdot 5^7 = 10^7$, the decimal form will have exactly 7 zeros..." (Mathematical Error)
> *Output:* "(E) 10"

This case suggests that even when the model does not explicitly "discuss" the peers, the latent representation of the interaction history acts as a distractor or noise, inducing reasoning drift and leading to conformity failures even when peers are correct.

## F    ADDITIONAL ANALYSIS

### F.1 EFFECT OF MCQ VS. OPEN-ENDED FORMATS

KAIROS adopts a multiple-choice (MCQ) formulation to enable precise and reproducible control of social pressure. Manipulations such as selecting the model's most plausible incorrect answer (e.g., in the *oppose-hard* condition) require access to a discrete belief distribution, which is well-defined in MCQs but unstable in open-ended generation. The MCQ format also supports exact-match evaluation, avoiding the variability inherent in LLM-as-judge scoring.

To assess whether this structure attenuates social influence, we conducted parallel experiments using open-ended generation on the reasoning subsets. The results show that open-ended answering is not only substantially harder—leading to sharp drops in original accuracy—but also markedly more sensitive to peer pressure. Without explicit options as an anchor, the model's uncertainty expands,

| Model Setup | Original | KAIROS | O–K $\Delta$ |
|---|---|---|---|
| Qwen2.5-3B (MCQ) | 47.93% | 48.77% | +2% |
| Qwen2.5-3B (Open) | 15.64% | 10.28% | -34% |
| Qwen2.5-7B (MCQ) | 58.50% | 52.27% | -11% |
| Qwen2.5-7B (Open) | 19.84% | 17.39% | -12% |
| Qwen2.5-14B (MCQ) | 64.00% | 58.43% | -9% |
| Qwen2.5-14B (Open) | 26.56% | 20.09% | -24% |

Table 5: Comparison of model performance under MCQ and open-ended formats. Open-ended generation substantially reduces base accuracy and amplifies susceptibility to peer influence (O–K), indicating that the MCQ setting provides a conservative lower bound on social vulnerability.

| Model Setup | Original Acc | KAIROS Acc | O–K |
|---|---|---|---|
| Qwen2.5-3B-Instruct (Standard) | 47.93% | 48.77% | +2% |
| Qwen2.5-3B-Instruct (High Conf.) | 48.47% | 47.40% | -2% |
| Qwen2.5-3B-Instruct (Low Conf.) | 48.33% | 47.03% | -3% |
| Qwen2.5-7B-Instruct (Standard) | 58.50% | 52.27% | -11% |
| Qwen2.5-7B-Instruct (High Conf.) | 58.50% | 52.73% | -10% |
| Qwen2.5-7B-Instruct (Low Conf.) | 58.40% | 52.23% | -11% |
| Qwen2.5-14B-Instruct (Standard) | 64.00% | 58.43% | -9% |
| Qwen2.5-14B-Instruct (High Conf.) | 63.90% | 57.70% | -10% |
| Qwen2.5-14B-Instruct (Low Conf.) | 64.07% | 58.20% | -9% |

Table 6: Effect of historical question confidence on susceptibility to peer influence. Conformity rates (O–K) remain stable across high- and low-confidence histories, indicating that intrinsic difficulty does not modulate social susceptibility.

giving peer responses a stronger influence on its generation trajectory. Quantitatively, the amplification of conformity is consistent across model sizes: Qwen2.5–3B shifts from a slight gain in MCQs (+2%) to a dramatic $-34\%$ drop in open-ended form, while Qwen2.5–14B declines $-9\%$ under MCQs but $-24\%$ in the open-ended setting.

Overall, these findings show that the MCQ setting does not artificially inflate conformity; rather, it provides a conservative estimate. Once the structural anchor of explicit options is removed, peer influence becomes considerably stronger. Full details and qualitative examples are included in the appendix.

### F.2 ANALYSIS OF ACTIVE HISTORY AND CONFIDENCE EFFECTS

**Does historical confidence modulate social influence?** We first test whether a model's susceptibility to peer influence is confounded by its confidence on historical questions. Using entropy as a proxy for uncertainty, we construct high- and low-confidence history sets while holding all peer conditions fixed. As shown in Table 6, conformity rates remain nearly identical across confidence levels for Qwen2.5–3B/7B/14B. This indicates that models are not responding to the intrinsic difficulty of prior questions; rather, they attend primarily to the social outcome of those interactions (i.e., whether peers appeared reliable). Historical confidence therefore does not meaningfully modulate social susceptibility.

**Does rapport reflect history content or merely context length?** To determine whether rapport effects stem from genuine interpretation of history rather than extended context, we introduce a *Masked History* control where all past peer responses are replaced with "$\star\star\star$". The model thus observes the presence and length of the history but receives no information about peer reliability. A representative snippet is shown below:

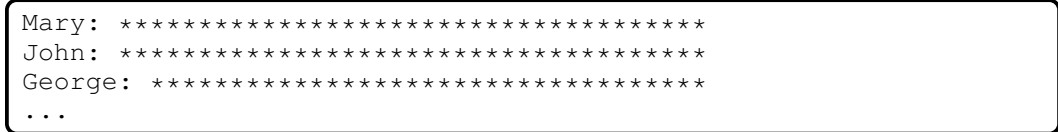

```
Mary:   *********************************
John:   *********************************
George: *********************************
...
```

This serves as a "default trust" baseline. As shown in Table 7, Qwen2.5–7B retains correct answers only 29.61% of the time under the *oppose-hard* condition—demonstrating strong blind conformity when no reliability cues are available. Introducing real interaction history, even fully unreliable history (0% rapport), substantially increases resistance (e.g., from 29.61% to 66.07%), with behavior scaling smoothly across rapport levels. Combined with the confidence results in Table 6, this confirms that interaction history functions as an explicit trust signal rather than incidental context or noise.

| History Condition | Support | Oppose-Hard | Oppose-Easy |
|---|---|---|---|
| Masked (No Rapport Info) | 97.95% | 29.61% | 34.37% |
| Rapport 100% | 97.46% | 42.74% | 41.96% |
| Rapport 75% | 93.52% | 52.14% | 55.74% |
| Rapport 50% | 88.80% | 56.10% | 63.30% |
| Rapport 25% | 87.29% | 65.22% | 72.52% |
| Rapport 0% | 81.03% | 66.07% | 66.67% |

Table 7: Correct-to-Correct (C→C) transition rates for Qwen2.5–7B. Masked history reveals a strong default-conformity baseline. Real history systematically modulates resistance based on peer reliability, demonstrating that models interpret interaction history as a trust signal.

## G   COMPREHENSIVE EVALUATION AND ANALYSIS RESULTS

We summarises the full evaluation results under KAIROS along with transition analyses for all model groups. Tables 8 to 11 report per-dataset performance across the four evaluation domains—*Reasoning*, *Knowledge*, *Social*, and *Creativity*. Each table includes Original and KAIROS accuracies, with bold entries indicating the per-dataset extrema (max/min) of O–KΔ.

Figures 5 to 9 present the transition analysis for all evaluated model families. Each figure visualises how prediction outcomes change under varying peer rapport levels and peer behaviours (SUPPORT, OPPOSEEASY, OPPOSEHARD), summarised across the four correctness transitions (Correct→Correct, Correct→Wrong, Wrong→Correct, Wrong→Wrong). Bubble size reflects transition frequency and colour intensity denotes confidence.

| Model | Type | BBH (n=334) | | | LiveCodeBench (n=333) | | | MATH-500 (n=333) | | |
|---|---|---|---|---|---|---|---|---|---|---|
| | | Original acc (↑) | KAIROS acc (↑) | O–K Δ (↓) | Original acc (↑) | KAIROS acc (↑) | O–K Δ (↓) | Original acc (↑) | KAIROS acc (↑) | O–K Δ (↓) |
| Qwen2.5-3B | Base | 47.60% | 47.01% | -1.3% | 40.24% | 42.04% | +4.5% | 32.13% | 36.64% | +14.0% |
| | Empowered | 57.49% | 43.11% | -25.0% | 53.45% | 41.44% | -22.5% | 63.36% | 34.23% | -46.0% |
| | Reflected | 47.60% | 47.60% | +0.0% | 40.24% | 42.04% | +4.5% | 32.13% | 31.53% | -1.9% |
| | SFT | 58.08% | 46.71% | -19.6% | 43.24% | 33.03% | -23.6% | 27.03% | 23.12% | -14.4% |
| | GRPO-MAS-DS-DR | 54.79% | 50.60% | -7.6% | 57.06% | 51.65% | -9.5% | 55.56% | 53.75% | -3.2% |
| | GRPO-MAS-NS-OR | 65.27% | 52.69% | -19.3% | 63.06% | 52.55% | -16.7% | 64.86% | 60.06% | -7.4% |
| | GRPO-nonMAS-DS-DR | 63.47% | 49.70% | -21.7% | 62.46% | 54.35% | -13.0% | 56.46% | 40.24% | -28.7% |
| | GRPO-nonMAS-NS-OR | 68.56% | 51.20% | -25.3% | 63.36% | 52.55% | -17.1% | 62.76% | 60.06% | -4.3% |
| | GRPO-MAS-DS-DR-LConf | 55.69% | 47.31% | -15.0% | 49.55% | 46.25% | -6.7% | 51.65% | 48.95% | -5.2% |
| | GRPO-MAS-DS-DR-LCorr | 53.89% | 47.01% | -12.8% | 49.85% | 45.95% | -7.8% | 58.56% | 55.26% | -5.6% |
| | GRPO-MAS-NS-OR-LConf | 61.68% | 46.41% | -24.8% | 55.86% | 48.05% | -14.0% | 65.77% | 57.36% | -12.8% |
| | GRPO-MAS-NS-OR-LCorr | 61.98% | 45.21% | -27.1% | 61.56% | 47.15% | -23.4% | 64.86% | 57.96% | -10.7% |
| Qwen2.5-7B | Base | 58.68% | 51.20% | -12.8% | 49.25% | 46.85% | -4.9% | 31.23% | 39.04% | +25.0% |
| | Empowered | 60.78% | 50.00% | -17.7% | 76.58% | 50.75% | -33.7% | 66.97% | 44.44% | -33.6% |
| | Reflected | 58.68% | 49.10% | -16.3% | 49.25% | 50.45% | +2.4% | 31.23% | 55.26% | +76.9% |
| | SFT | 64.97% | 43.71% | -32.7% | 54.65% | 39.34% | -28.0% | 28.83% | 36.34% | +26.0% |
| | GRPO-MAS-DS-DR | 71.26% | 66.77% | -6.3% | 62.46% | 61.56% | -1.4% | 66.67% | 57.36% | -14.0% |
| | GRPO-MAS-NS-OR | 76.65% | 74.25% | -3.1% | 73.57% | 70.57% | -4.1% | 71.17% | 73.87% | +3.8% |
| | GRPO-nonMAS-DS-DR | 71.26% | 59.58% | -16.4% | 63.96% | 62.46% | -2.3% | 62.16% | 57.66% | -7.2% |
| | GRPO-nonMAS-NS-OR | 83.53% | 65.27% | -21.9% | 78.68% | 64.86% | -17.6% | 81.38% | 67.57% | -17.0% |
| | GRPO-MAS-DS-DR-LConf | 65.27% | 56.89% | -12.8% | 51.65% | 56.46% | +9.3% | 54.65% | 52.85% | -3.3% |
| | GRPO-MAS-DS-DR-LCorr | 62.57% | 49.10% | -21.5% | 59.16% | 47.75% | -19.3% | 59.76% | 54.95% | -8.0% |
| | GRPO-MAS-NS-OR-LConf | 72.16% | 56.29% | -22.0% | 75.98% | 58.26% | -23.3% | 76.28% | 68.47% | -10.2% |
| | GRPO-MAS-NS-OR-LCorr | 72.46% | 52.40% | -27.7% | 75.38% | 61.26% | -18.7% | 76.58% | 69.67% | -9.0% |
| Qwen2.5-14B | Base | 62.57% | 59.28% | -5.3% | 62.46% | 55.56% | -11.1% | 40.24% | 43.84% | +8.9% |
| | Empowered | 62.87% | 61.38% | -2.4% | 71.17% | 59.16% | -16.9% | 75.08% | 72.97% | -2.8% |
| | Reflected | 62.57% | 59.28% | -5.3% | 62.46% | 54.35% | -13.0% | 40.24% | 49.55% | +23.1% |
| | SFT | 70.96% | 44.91% | -36.7% | 68.77% | 36.04% | **−47.6%** | 35.14% | 36.34% | +3.4% |
| | GRPO-MAS-DS-DR | 82.04% | 82.63% | +0.7% | 78.68% | 66.67% | -15.3% | 72.67% | 69.07% | -5.0% |
| | GRPO-MAS-NS-OR | 86.53% | 82.34% | -4.8% | 83.48% | 73.57% | -11.9% | 78.08% | 74.77% | -4.2% |
| | GRPO-nonMAS-DS-DR | 80.24% | 72.16% | -10.1% | 75.38% | 65.17% | -13.5% | 66.37% | 66.67% | +0.5% |
| | GRPO-nonMAS-NS-OR | 86.53% | 73.05% | -15.6% | 86.49% | 70.27% | -18.8% | 80.48% | 72.37% | -10.1% |
| | GRPO-MAS-DS-DR-LConf | 74.85% | 59.58% | -20.4% | 69.07% | 59.76% | -13.5% | 60.96% | 60.06% | -1.5% |
| | GRPO-MAS-DS-DR-LCorr | 75.45% | 44.91% | -40.5% | 66.37% | 47.15% | -29.0% | 62.16% | 51.95% | -16.4% |
| | GRPO-MAS-NS-OR-LConf | 78.44% | 61.98% | -21.0% | 81.98% | 66.07% | -19.4% | 79.28% | 66.07% | -16.7% |
| | GRPO-MAS-NS-OR-LCorr | 78.44% | 54.19% | -30.9% | 77.18% | 65.47% | -15.2% | 75.68% | 71.17% | -5.9% |
| Llama3.2-3B | Base | 49.70% | 46.71% | -6.0% | 30.03% | 24.02% | -20.0% | 22.52% | 26.73% | +18.7% |
| | Empowered | 49.40% | 44.61% | -9.7% | 29.13% | 25.53% | -12.4% | 21.62% | 22.52% | +4.2% |
| | Reflected | 49.70% | 35.93% | -27.7% | 30.03% | 29.13% | -3.0% | 22.52% | 21.02% | -6.7% |
| | SFT | 55.09% | 38.62% | -29.9% | 34.53% | 35.44% | +2.6% | 18.92% | 35.14% | +**85.7**% |
| | GRPO-MAS-DS-DR | 53.59% | 44.01% | -17.9% | 27.33% | 34.83% | +27.5% | 27.63% | 30.93% | +11.9% |
| | GRPO-MAS-NS-OR | 61.38% | 51.80% | -15.6% | 33.63% | 39.64% | +17.9% | 39.34% | 42.64% | +8.4% |
| | GRPO-nonMAS-DS-DR | 62.87% | 44.31% | -29.5% | 33.03% | 30.93% | -6.4% | 39.34% | 32.73% | -16.8% |
| | GRPO-nonMAS-NS-OR | 60.78% | 44.91% | -26.1% | 43.54% | 42.64% | -2.1% | 47.45% | 38.74% | -18.4% |
| | GRPO-MAS-DS-DR-LConf | 55.99% | 42.22% | -24.6% | 29.13% | 34.53% | +18.6% | 30.93% | 28.23% | -8.7% |
| | GRPO-MAS-DS-DR-LCorr | 54.79% | 46.11% | -15.8% | 26.73% | 34.83% | +**30.3**% | 28.53% | 33.63% | +17.9% |
| | GRPO-MAS-NS-OR-LConf | 63.77% | 52.69% | -17.4% | 31.53% | 38.14% | +21.0% | 46.25% | 40.54% | -12.3% |
| | GRPO-MAS-NS-OR-LCorr | 63.77% | 52.69% | -17.4% | 31.53% | 38.14% | +21.0% | 46.25% | 40.54% | -12.3% |
| Llama3.1-8B | Base | 55.39% | 47.31% | -14.6% | 32.43% | 37.84% | +16.7% | 32.13% | 29.43% | -8.4% |
| | Empowered | 59.88% | 46.71% | -22.0% | 46.25% | 40.54% | -12.3% | 54.35% | 28.83% | −**47.0**% |
| | Reflected | 55.39% | 40.72% | -26.5% | 32.43% | 33.33% | +2.8% | 32.13% | 22.52% | -29.9% |
| | SFT | 63.47% | 36.53% | −**42.5**% | 49.55% | 39.34% | -20.6% | 21.02% | 33.93% | +61.4% |
| | GRPO-MAS-DS-DR | 65.57% | 59.28% | -9.6% | 44.44% | 40.84% | -8.1% | 43.84% | 38.14% | -13.0% |
| | GRPO-MAS-NS-OR | 69.16% | 62.87% | -9.1% | 60.36% | 43.54% | -27.9% | 48.95% | 37.24% | -23.9% |
| | GRPO-nonMAS-DS-DR | 64.37% | 47.60% | -26.1% | 47.15% | 38.74% | -17.8% | 34.53% | 33.93% | -1.7% |
| | GRPO-nonMAS-NS-OR | 74.25% | 61.98% | -16.5% | 58.86% | 47.45% | -19.4% | 48.05% | 39.94% | -16.9% |
| | GRPO-MAS-DS-DR-LConf | 65.27% | 50.00% | -23.4% | 40.54% | 35.74% | -11.8% | 33.33% | 37.84% | +13.5% |
| | GRPO-MAS-DS-DR-LCorr | 69.16% | 43.41% | -37.2% | 44.44% | 41.14% | -7.4% | 46.25% | 40.54% | -12.3% |
| | GRPO-MAS-NS-OR-LConf | 68.86% | 54.79% | -20.4% | 54.05% | 38.14% | -29.5% | 50.45% | 42.04% | -16.7% |
| | GRPO-MAS-NS-OR-LCorr | 71.86% | 56.59% | -21.2% | 49.85% | 38.74% | -22.3% | 48.65% | 37.84% | -22.2% |
| Llama3.3-70B | Base | 64.67% | 67.37% | +4.2% | 57.66% | 59.46% | +3.1% | 35.14% | 38.74% | +10.2% |
| | Empowered | 64.37% | 69.49% | **+8.0%** | 57.66% | 58.56% | +1.6% | 36.34% | 39.94% | +9.9% |
| | Reflected | 64.67% | 66.77% | +3.2% | 57.66% | 55.56% | -3.6% | 35.14% | 31.23% | -11.1% |
| Qwen2.5-32B | Base | 69.16% | 67.96% | -1.7% | 68.17% | 64.86% | -4.9% | 42.34% | 43.84% | +3.5% |
| | Empowered | 67.07% | 63.77% | -4.9% | 69.97% | 63.06% | -9.9% | 55.26% | 46.55% | -15.8% |
| | Reflected | 69.16% | 61.98% | -10.4% | 68.17% | 60.36% | -11.5% | 42.34% | 44.74% | +5.7% |
| Qwen2.5-72B | Base | 68.26% | 66.17% | -3.1% | 64.56% | 64.26% | -0.5% | 42.04% | 43.54% | +3.6% |
| | Empowered | 68.86% | 67.66% | -1.7% | 64.56% | 66.37% | +2.8% | 42.94% | 45.65% | +6.3% |
| | Reflected | 68.26% | 65.57% | -3.9% | 64.56% | 63.36% | -1.9% | 42.04% | 45.95% | +9.3% |
| GPT-OSS-120B | Base | 89.22% | 82.93% | -7.0% | 98.50% | 94.89% | -4.6% | 98.50% | 92.49% | -6.1% |
| | Empowered | 89.22% | 87.72% | -1.7% | 98.80% | 96.40% | -2.4% | 98.50% | 93.09% | -5.5% |
| | Reflected | 89.22% | 88.92% | -0.3% | 98.50% | 95.50% | -3.0% | 98.50% | 91.89% | -6.7% |
| Gemini-2.5-Pro | Base | 92.81% | 80.84% | -12.9% | 97.60% | 91.89% | -5.9% | 99.10% | 95.20% | -3.9% |
| | Empowered | 90.12% | 91.92% | +2.0% | 97.00% | 97.30% | +0.3% | 99.40% | 99.40% | +0.0% |
| | Reflected | 92.81% | 93.11% | +0.3% | 97.60% | 97.90% | +0.3% | 99.10% | 99.70% | +0.6% |
| GPT-5 | Base | 92.81% | 91.32% | -1.6% | 99.10% | 98.80% | -0.3% | 98.80% | 98.50% | -0.3% |
| | Empowered | 91.92% | 92.81% | +1.0% | 99.10% | 98.80% | -0.3% | 98.80% | 99.10% | +0.3% |
| | Reflected | 92.81% | 93.11% | +0.3% | 99.10% | 98.80% | -0.3% | 98.80% | 99.10% | +0.3% |

Table 8: Overall results for the **Reasoning** category under KAIROS. **Bold** numbers mark per-dataset extreme (max/min) O–K Δ.

| Model | Type | MMLU-Pro (n=335) | | | TruthfulQA (n=333) | | |
|---|---|---|---|---|---|---|---|
| | | Original acc (↑) | KAIROS acc (↑) | O–K Δ (↓) | Original acc (↑) | KAIROS acc (↑) | O–K Δ (↓) |
| Qwen2.5-3B | Base | 54.33% | 46.87% | -13.7% | 57.66% | 67.57% | +17.2% |
| | Empowered | 61.19% | 46.27% | -24.4% | 56.46% | 67.87% | +20.2% |
| | Reflected | 54.33% | 48.36% | -11.0% | 57.66% | 60.66% | +5.2% |
| | SFT | 48.96% | 41.19% | -15.8% | 45.95% | 43.24% | -5.9% |
| | GRPO-MAS-DS-DR | 61.49% | 52.84% | -14.1% | 54.65% | 51.65% | -5.5% |
| | GRPO-MAS-NS-OR | 65.67% | 58.81% | -10.5% | 57.96% | 57.06% | -1.6% |
| | GRPO-nonMAS-DS-DR | 62.39% | 52.54% | -15.8% | 56.16% | 55.86% | -0.5% |
| | GRPO-nonMAS-NS-OR | 64.78% | 53.73% | -17.0% | 55.26% | 46.55% | -15.8% |
| | GRPO-MAS-DS-DR-LConf | 59.10% | 50.75% | -14.1% | 54.95% | 50.15% | -8.7% |
| | GRPO-MAS-DS-DR-LCorr | 65.97% | 47.16% | -28.5% | 54.35% | 41.44% | -23.8% |
| | GRPO-MAS-NS-OR-LConf | 68.06% | 52.24% | -23.2% | 57.06% | 50.15% | -12.1% |
| | GRPO-MAS-NS-OR-LCorr | 66.57% | 45.67% | -31.4% | 55.86% | 43.54% | -22.0% |
| Qwen2.5-7B | Base | 60.00% | 51.04% | -14.9% | 67.27% | 63.36% | -5.8% |
| | Empowered | 63.58% | 57.01% | -10.3% | 66.97% | 65.47% | -2.2% |
| | Reflected | 60.00% | 59.40% | -1.0% | 67.27% | 64.56% | -4.0% |
| | SFT | 54.93% | 41.19% | -25.0% | 52.55% | 46.55% | -11.4% |
| | GRPO-MAS-DS-DR | 68.06% | 63.58% | -6.6% | 63.66% | 58.86% | -7.5% |
| | GRPO-MAS-NS-OR | 71.94% | 68.36% | -5.0% | 60.36% | 53.15% | -11.9% |
| | GRPO-nonMAS-DS-DR | 68.96% | 60.00% | -13.0% | 59.16% | 58.56% | -1.0% |
| | GRPO-nonMAS-NS-OR | 76.42% | 62.69% | -18.0% | 59.76% | 53.15% | -11.1% |
| | GRPO-MAS-DS-DR-LConf | 65.37% | 57.01% | -12.8% | 64.56% | 52.85% | -18.1% |
| | GRPO-MAS-DS-DR-LCorr | 68.96% | 52.54% | -23.8% | 62.46% | 44.44% | -28.9% |
| | GRPO-MAS-NS-OR-LConf | 74.03% | 61.19% | -17.3% | 66.97% | 55.26% | -17.5% |
| | GRPO-MAS-NS-OR-LCorr | 74.03% | 57.61% | -22.2% | 66.07% | 41.74% | -36.8% |
| Qwen2.5-14B | Base | 67.16% | 61.19% | -8.9% | 73.57% | 77.48% | +5.3% |
| | Empowered | 69.55% | 63.88% | -8.2% | 76.28% | 77.78% | +2.0% |
| | Reflected | 67.16% | 66.27% | -1.3% | 73.57% | 72.07% | -2.0% |
| | SFT | 64.48% | 42.39% | −**34.3**% | 71.17% | 58.26% | -18.1% |
| | GRPO-MAS-DS-DR | 77.31% | 68.96% | -10.8% | 72.97% | 69.37% | -4.9% |
| | GRPO-MAS-NS-OR | 78.51% | 67.16% | -14.4% | 74.47% | 70.87% | -4.8% |
| | GRPO-nonMAS-DS-DR | 77.61% | 65.07% | -16.2% | 69.37% | 66.97% | -3.5% |
| | GRPO-nonMAS-NS-OR | 78.81% | 68.66% | -12.9% | 71.77% | 66.07% | -8.0% |
| | GRPO-MAS-DS-DR-LConf | 72.24% | 63.88% | -11.6% | 73.27% | 72.97% | -0.4% |
| | GRPO-MAS-DS-DR-LCorr | 71.64% | 49.55% | -30.8% | 69.37% | 49.85% | -28.1% |
| | GRPO-MAS-NS-OR-LConf | 79.40% | 61.79% | -22.2% | 72.07% | 59.46% | -17.5% |
| | GRPO-MAS-NS-OR-LCorr | 80.00% | 54.93% | -31.3% | 63.36% | 37.84% | −**40.3**% |
| Llama3.2-3B | Base | 48.06% | 38.21% | -20.5% | 52.25% | 57.96% | +10.9% |
| | Empowered | 47.46% | 39.70% | -16.4% | 56.16% | 61.56% | +9.6% |
| | Reflected | 48.06% | 34.03% | -29.2% | 52.25% | 37.24% | -28.7% |
| | SFT | 38.81% | 34.33% | -11.5% | 43.24% | 36.04% | -16.7% |
| | GRPO-MAS-DS-DR | 50.15% | 44.18% | -11.9% | 54.65% | 48.35% | -11.5% |
| | GRPO-MAS-NS-OR | 53.73% | 48.36% | -10.0% | 49.55% | 50.75% | +2.4% |
| | GRPO-nonMAS-DS-DR | 52.84% | 40.30% | -23.7% | 57.06% | 49.25% | -13.7% |
| | GRPO-nonMAS-NS-OR | 57.91% | 45.97% | -20.6% | 55.26% | 55.86% | +1.1% |
| | GRPO-MAS-DS-DR-LConf | 50.15% | 38.21% | -23.8% | 54.95% | 54.95% | +0.0% |
| | GRPO-MAS-DS-DR-LCorr | 53.13% | 46.27% | -12.9% | 55.26% | 46.55% | -15.8% |
| | GRPO-MAS-NS-OR-LConf | 55.82% | 45.37% | -18.7% | 50.75% | 50.15% | -1.2% |
| | GRPO-MAS-NS-OR-LCorr | 55.82% | 45.37% | -18.7% | 50.75% | 50.15% | -1.2% |
| Llama3.1-8B | Base | 60.60% | 48.66% | -19.7% | 58.26% | 69.37% | +19.1% |
| | Empowered | 64.48% | 51.04% | -20.8% | 62.16% | 68.77% | +10.6% |
| | Reflected | 60.60% | 48.66% | -19.7% | 58.26% | 52.25% | -10.3% |
| | SFT | 44.48% | 37.01% | -16.8% | 32.13% | 34.53% | +7.5% |
| | GRPO-MAS-DS-DR | 62.09% | 52.84% | -14.9% | 57.36% | 57.96% | +1.0% |
| | GRPO-MAS-NS-OR | 65.37% | 51.94% | -20.5% | 53.75% | 58.26% | +8.4% |
| | GRPO-nonMAS-DS-DR | 60.00% | 49.55% | -17.4% | 56.76% | 61.56% | +8.5% |
| | GRPO-nonMAS-NS-OR | 65.97% | 52.84% | -19.9% | 43.54% | 52.85% | +**21.4**% |
| | GRPO-MAS-DS-DR-LConf | 61.19% | 49.25% | -19.5% | 58.86% | 60.96% | +3.6% |
| | GRPO-MAS-DS-DR-LCorr | 63.58% | 42.99% | -32.4% | 56.76% | 52.55% | -7.4% |
| | GRPO-MAS-NS-OR-LConf | 64.48% | 53.73% | -16.7% | 54.35% | 48.95% | -10.0% |
| | GRPO-MAS-NS-OR-LCorr | 64.48% | 54.33% | -15.7% | 57.06% | 56.76% | -0.5% |
| Llama3.3-70B | Base | 67.76% | 66.87% | -1.3% | 73.57% | 74.77% | +1.6% |
| | Empowered | 68.66% | 68.66% | +0.0% | 77.18% | 75.08% | -2.7% |
| | Reflected | 67.76% | 64.78% | -4.4% | 73.57% | 75.08% | +2.1% |
| Qwen2.5-32B | Base | 73.13% | 69.25% | -5.3% | 78.68% | 79.58% | +1.1% |
| | Empowered | 74.93% | 69.25% | -7.6% | 81.08% | 81.98% | +1.1% |
| | Reflected | 73.13% | 69.25% | -5.3% | 78.68% | 77.48% | -1.5% |
| Qwen2.5-72B | Base | 72.24% | 71.34% | -1.2% | 81.38% | 83.48% | +2.6% |
| | Empowered | 72.84% | 73.73% | +**1.2**% | 82.58% | 84.98% | +2.9% |
| | Reflected | 72.24% | 71.64% | -0.8% | 81.38% | 78.08% | -4.1% |
| GPT-OSS-120B | Base | 88.66% | 84.78% | -4.4% | 79.28% | 77.78% | -1.9% |
| | Empowered | 89.85% | 86.27% | -4.0% | 81.68% | 81.98% | +0.4% |
| | Reflected | 88.66% | 85.97% | -3.0% | 79.28% | 84.98% | +7.2% |
| Gemini-2.5-Pro | Base | 92.54% | 83.28% | -10.0% | 89.79% | 85.29% | -5.0% |
| | Empowered | 91.94% | 91.04% | -1.0% | 90.69% | 92.49% | +2.0% |
| | Reflected | 92.54% | 91.34% | -1.3% | 89.79% | 92.79% | +3.3% |
| GPT-5 | Base | 92.84% | 92.84% | +0.0% | 81.68% | 87.99% | +7.7% |
| | Empowered | 91.34% | 91.64% | +0.3% | 84.38% | 90.09% | +6.8% |
| | Reflected | 92.84% | 92.24% | -0.6% | 81.68% | 89.49% | +9.6% |

Table 9: Overall results for the **Knowledge** category under KAIROS. **Bold** numbers mark per-dataset extreme (max/min) O–K Δ.

| Model | Type | CommonSenseQA (n=333) | | | Social IQa (n=333) | | |
|---|---|---|---|---|---|---|---|
| | | Original acc (↑) | KAIROS acc (↑) | O–K Δ (↓) | Original acc (↑) | KAIROS acc (↑) | O–K Δ (↓) |
| Qwen2.5-3B | Base | 57.36% | 59.16% | +3.1% | 70.27% | 64.56% | -8.1% |
| | Empowered | 58.86% | 59.16% | +0.5% | 70.87% | 63.66% | -10.2% |
| | Reflected | 57.36% | 58.26% | +1.6% | 70.27% | 56.46% | -19.7% |
| | SFT | 52.85% | 60.36% | +14.2% | 70.27% | 60.36% | -14.1% |
| | GRPO-MAS-DS-DR | 62.16% | 58.26% | -6.3% | 68.47% | 64.56% | -5.7% |
| | GRPO-MAS-NS-OR | 65.17% | 63.96% | -1.8% | 71.47% | 62.46% | -12.6% |
| | GRPO-nonMAS-DS-DR | 60.96% | 54.65% | -10.4% | 72.97% | 63.06% | -13.6% |
| | GRPO-nonMAS-NS-OR | 63.36% | 60.36% | -4.7% | 72.37% | 61.56% | -14.9% |
| | GRPO-MAS-DS-DR-LConf | 61.56% | 58.86% | -4.4% | 70.87% | 60.96% | -14.0% |
| | GRPO-MAS-DS-DR-LCorr | 61.56% | 58.86% | -4.4% | 68.77% | 45.65% | -33.6% |
| | GRPO-MAS-NS-OR-LConf | 61.86% | 61.86% | +0.0% | 72.97% | 56.16% | -23.1% |
| | GRPO-MAS-NS-OR-LCorr | 64.26% | 55.56% | -13.6% | 72.37% | 43.54% | -39.8% |
| Qwen2.5-7B | Base | 68.17% | 56.76% | -16.7% | 74.17% | 68.77% | -7.3% |
| | Empowered | 65.77% | 57.36% | -12.8% | 75.08% | 69.97% | -6.8% |
| | Reflected | 68.17% | 61.56% | -9.7% | 74.17% | 72.97% | -1.6% |
| | SFT | 60.96% | 44.14% | -27.6% | 68.17% | 50.15% | -26.4% |
| | GRPO-MAS-DS-DR | 69.07% | 67.87% | -1.7% | 71.47% | 68.47% | -4.2% |
| | GRPO-MAS-NS-OR | 75.08% | 59.16% | -21.2% | 74.17% | 66.07% | -10.9% |
| | GRPO-nonMAS-DS-DR | 68.47% | 65.17% | -4.8% | 71.17% | 61.56% | -13.5% |
| | GRPO-nonMAS-NS-OR | 76.28% | 57.66% | -24.4% | 75.98% | 54.05% | -28.8% |
| | GRPO-MAS-DS-DR-LConf | 69.97% | 55.56% | -20.6% | 72.97% | 56.16% | -23.1% |
| | GRPO-MAS-DS-DR-LCorr | 71.47% | 57.36% | -19.7% | 73.27% | 45.65% | -37.7% |
| | GRPO-MAS-NS-OR-LConf | 73.87% | 63.96% | -13.4% | 76.28% | 59.76% | -21.7% |
| | GRPO-MAS-NS-OR-LCorr | 76.88% | 49.55% | -35.5% | 74.77% | 42.04% | −43.8% |
| Qwen2.5-14B | Base | 69.37% | 64.56% | -6.9% | 74.47% | 66.97% | -10.1% |
| | Empowered | 70.57% | 69.37% | -1.7% | 73.57% | 69.07% | -6.1% |
| | Reflected | 69.37% | 64.86% | -6.5% | 74.47% | 69.07% | -7.3% |
| | SFT | 68.47% | 52.85% | -22.8% | 70.27% | 48.05% | -31.6% |
| | GRPO-MAS-DS-DR | 83.18% | 77.18% | -7.2% | 75.08% | 71.77% | -4.4% |
| | GRPO-MAS-NS-OR | 81.38% | 78.08% | -4.1% | 73.27% | 72.07% | -1.6% |
| | GRPO-nonMAS-DS-DR | 80.48% | 63.96% | -20.5% | 75.68% | 65.77% | -13.1% |
| | GRPO-nonMAS-NS-OR | 82.28% | 63.96% | -22.3% | 76.58% | 64.56% | -15.7% |
| | GRPO-MAS-DS-DR-LConf | 76.88% | 68.77% | -10.5% | 75.98% | 61.56% | -19.0% |
| | GRPO-MAS-DS-DR-LCorr | 76.28% | 48.35% | -36.6% | 72.67% | 42.04% | -42.1% |
| | GRPO-MAS-NS-OR-LConf | 81.08% | 62.46% | -23.0% | 76.28% | 56.16% | -26.4% |
| | GRPO-MAS-NS-OR-LCorr | 82.58% | 48.35% | −41.5% | 73.57% | 44.74% | -39.2% |
| Llama3.2-3B | Base | 57.66% | 50.75% | -12.0% | 63.96% | 50.75% | -20.7% |
| | Empowered | 59.76% | 51.65% | -13.6% | 64.86% | 55.86% | -13.9% |
| | Reflected | 57.66% | 49.55% | -14.1% | 63.96% | 47.15% | -26.3% |
| | SFT | 52.55% | 48.65% | -7.4% | 57.66% | 38.74% | -32.8% |
| | GRPO-MAS-DS-DR | 59.16% | 49.85% | -15.7% | 67.87% | 55.26% | -18.6% |
| | GRPO-MAS-NS-OR | 62.16% | 60.36% | -2.9% | 72.07% | 60.96% | -15.4% |
| | GRPO-nonMAS-DS-DR | 59.46% | 49.85% | -16.2% | 67.87% | 56.46% | -16.8% |
| | GRPO-nonMAS-NS-OR | 57.96% | 57.36% | -1.0% | 73.27% | 56.76% | -22.5% |
| | GRPO-MAS-DS-DR-LConf | 57.66% | 51.95% | -9.9% | 67.27% | 52.25% | -22.3% |
| | GRPO-MAS-DS-DR-LCorr | 57.06% | 50.75% | -11.1% | 67.87% | 45.35% | -33.2% |
| | GRPO-MAS-NS-OR-LConf | 65.47% | 56.46% | -13.8% | 71.47% | 53.45% | -25.2% |
| | GRPO-MAS-NS-OR-LCorr | 65.47% | 56.46% | -13.8% | 71.47% | 53.45% | -25.2% |
| Llama3.1-8B | Base | 68.17% | 60.66% | -11.0% | 73.27% | 62.76% | -14.3% |
| | Empowered | 65.47% | 59.76% | -8.7% | 72.07% | 65.47% | -9.2% |
| | Reflected | 68.17% | 50.15% | -26.4% | 73.27% | 54.95% | -25.0% |
| | SFT | 53.15% | 53.45% | +0.6% | 65.77% | 43.54% | -33.8% |
| | GRPO-MAS-DS-DR | 63.66% | 61.26% | -3.8% | 72.07% | 62.76% | -12.9% |
| | GRPO-MAS-NS-OR | 66.37% | 63.96% | -3.6% | 72.37% | 66.37% | -8.3% |
| | GRPO-nonMAS-DS-DR | 67.27% | 55.86% | -17.0% | 75.08% | 56.76% | -24.4% |
| | GRPO-nonMAS-NS-OR | 69.67% | 66.07% | -5.2% | 73.57% | 57.36% | -22.0% |
| | GRPO-MAS-DS-DR-LConf | 65.77% | 64.56% | -1.8% | 71.77% | 57.36% | -20.1% |
| | GRPO-MAS-DS-DR-LCorr | 65.17% | 59.16% | -9.2% | 70.87% | 46.85% | -33.9% |
| | GRPO-MAS-NS-OR-LConf | 67.87% | 63.96% | -5.7% | 70.87% | 60.66% | -14.4% |
| | GRPO-MAS-NS-OR-LCorr | 72.37% | 63.96% | -11.6% | 72.07% | 58.56% | -18.7% |
| Llama3.3-70B | Base | 78.68% | 78.38% | -0.4% | 79.58% | 75.08% | -5.7% |
| | Empowered | 80.18% | 80.18% | +0.0% | 80.48% | 79.58% | -1.1% |
| | Reflected | 78.68% | 76.58% | -2.7% | 79.58% | 78.98% | -0.8% |
| Qwen2.5-32B | Base | 74.77% | 72.07% | -3.6% | 79.88% | 78.08% | -2.3% |
| | Empowered | 74.77% | 74.17% | -0.8% | 78.98% | 78.38% | -0.8% |
| | Reflected | 74.77% | 74.47% | -0.4% | 79.88% | 76.88% | -3.8% |
| Qwen2.5-72B | Base | 79.88% | 79.28% | -0.8% | 78.08% | 72.97% | -6.5% |
| | Empowered | 79.88% | 79.58% | -0.4% | 78.68% | 76.28% | -3.1% |
| | Reflected | 79.88% | 74.47% | -6.8% | 78.08% | 77.48% | -0.8% |
| GPT-OSS-120B | Base | 85.59% | 72.07% | -15.8% | 79.88% | 66.97% | -16.2% |
| | Empowered | 84.68% | 80.48% | -5.0% | 79.58% | 72.37% | -9.1% |
| | Reflected | 85.59% | 83.18% | -2.8% | 79.88% | 75.68% | -5.3% |
| Gemini-2.5-Pro | Base | 84.38% | 68.47% | -18.9% | 79.58% | 65.77% | -17.4% |
| | Empowered | 78.98% | 75.38% | -4.6% | 78.98% | 79.58% | +0.8% |
| | Reflected | 84.38% | 73.87% | -12.5% | 79.58% | 76.28% | -4.1% |
| GPT-5 | Base | 87.99% | 79.88% | -9.2% | 80.18% | 78.08% | -2.6% |
| | Empowered | 86.19% | 81.08% | -5.9% | 80.18% | 78.98% | -1.5% |
| | Reflected | 87.99% | 81.08% | -7.9% | 80.18% | 78.38% | -2.2% |

Table 10: Overall results for the **Social** category under KAIROS. **Bold** numbers mark per-dataset extreme (max/min) O–K Δ.

| Model | Type | BrainTeaser (n=333) | | | MacGyver (n=333) | | |
|---|---|---|---|---|---|---|---|
| | | Original acc (↑) | KAIROS acc (↑) | O–K Δ (↓) | Original acc (↑) | KAIROS acc (↑) | O–K Δ (↓) |
| Qwen2.5-3B | Base | 37.54% | 30.63% | -18.4% | 34.23% | 44.44% | +29.8% |
| | Empowered | 41.74% | 30.63% | -26.6% | 41.14% | 44.44% | +8.0% |
| | Reflected | 37.54% | 23.42% | -37.6% | 34.23% | 57.06% | +**66.7**% |
| | SFT | 54.35% | 43.24% | -20.4% | 51.05% | 72.37% | +41.8% |
| | GRPO-MAS-DS-DR | 35.14% | 33.63% | -4.3% | 44.14% | 48.35% | +9.5% |
| | GRPO-MAS-NS-OR | 39.04% | 41.44% | +6.1% | 62.46% | 72.07% | +15.4% |
| | GRPO-nonMAS-DS-DR | 36.94% | 35.44% | -4.1% | 46.55% | 55.56% | +19.4% |
| | GRPO-nonMAS-NS-OR | 44.44% | 34.23% | -23.0% | 69.67% | 63.96% | -8.2% |
| | GRPO-MAS-DS-DR-LConf | 28.53% | 28.53% | +0.0% | 40.84% | 47.75% | +16.9% |
| | GRPO-MAS-DS-DR-LCorr | 33.03% | 33.93% | +2.7% | 54.05% | 51.95% | -3.9% |
| | GRPO-MAS-NS-OR-LConf | 38.74% | 38.14% | -1.5% | 63.96% | 60.66% | -5.2% |
| | GRPO-MAS-NS-OR-LCorr | 35.44% | 31.53% | -11.0% | 59.16% | 57.06% | -3.5% |
| Qwen2.5-7B | Base | 42.94% | 47.45% | +10.5% | 74.77% | 45.95% | -38.6% |
| | Empowered | 44.14% | 46.25% | +4.8% | 71.77% | 45.35% | -36.8% |
| | Reflected | 42.94% | 42.94% | +0.0% | 74.77% | 41.74% | -44.2% |
| | SFT | 64.86% | 48.65% | -25.0% | 60.06% | 45.95% | -23.5% |
| | GRPO-MAS-DS-DR | 50.45% | 54.35% | +7.7% | 76.58% | 59.46% | -22.4% |
| | GRPO-MAS-NS-OR | 54.65% | 55.86% | +2.2% | 74.77% | 67.87% | -9.2% |
| | GRPO-nonMAS-DS-DR | 51.05% | 47.15% | -7.6% | 63.96% | 61.56% | -3.8% |
| | GRPO-nonMAS-NS-OR | 48.95% | 41.14% | -16.0% | 73.57% | 52.55% | -28.6% |
| | GRPO-MAS-DS-DR-LConf | 46.55% | 43.24% | -7.1% | 72.07% | 49.55% | -31.2% |
| | GRPO-MAS-DS-DR-LCorr | 43.24% | 42.04% | -2.8% | 68.77% | 54.95% | -20.1% |
| | GRPO-MAS-NS-OR-LConf | 46.85% | 38.74% | -17.3% | 74.47% | 49.55% | -33.5% |
| | GRPO-MAS-NS-OR-LCorr | 48.05% | 41.14% | -14.4% | 79.28% | 58.86% | -25.8% |
| Qwen2.5-14B | Base | 52.25% | 48.05% | -8.0% | 73.87% | 48.95% | -33.7% |
| | Empowered | 46.25% | 44.74% | -3.2% | 68.77% | 44.14% | -35.8% |
| | Reflected | 52.25% | 44.74% | -14.4% | 73.87% | 52.55% | -28.9% |
| | SFT | 70.87% | 60.96% | -14.0% | 67.87% | 59.76% | -11.9% |
| | GRPO-MAS-DS-DR | 61.56% | 59.76% | -2.9% | 76.58% | 60.36% | -21.2% |
| | GRPO-MAS-NS-OR | 60.36% | 63.36% | +5.0% | 71.77% | 61.26% | -14.6% |
| | GRPO-nonMAS-DS-DR | 57.96% | 53.45% | -7.8% | 72.07% | 43.24% | -40.0% |
| | GRPO-nonMAS-NS-OR | 60.66% | 54.35% | -10.4% | 74.17% | 55.86% | -24.7% |
| | GRPO-MAS-DS-DR-LConf | 56.16% | 52.55% | -6.4% | 71.47% | 48.65% | -31.9% |
| | GRPO-MAS-DS-DR-LCorr | 45.05% | 35.74% | -20.7% | 78.68% | 42.34% | -46.2% |
| | GRPO-MAS-NS-OR-LConf | 59.16% | 51.95% | -12.2% | 65.47% | 58.26% | -11.0% |
| | GRPO-MAS-NS-OR-LCorr | 58.56% | 45.05% | -23.1% | 67.57% | 47.75% | -29.3% |
| Llama3.2-3B | Base | 43.54% | 42.34% | -2.8% | 63.36% | 56.76% | -10.4% |
| | Empowered | 37.84% | 43.54% | +15.1% | 69.67% | 57.36% | -17.7% |
| | Reflected | 43.54% | 34.83% | -20.0% | 63.36% | 56.76% | -10.4% |
| | SFT | 48.05% | 37.54% | -21.9% | 54.65% | 48.95% | -10.4% |
| | GRPO-MAS-DS-DR | 49.25% | 43.84% | -11.0% | 69.97% | 63.36% | -9.4% |
| | GRPO-MAS-NS-OR | 51.95% | 45.35% | -12.7% | 77.78% | 61.56% | -20.9% |
| | GRPO-nonMAS-DS-DR | 49.85% | 39.94% | -19.9% | 77.18% | 60.96% | -21.0% |
| | GRPO-nonMAS-NS-OR | 49.55% | 43.54% | -12.1% | 77.78% | 66.37% | -14.7% |
| | GRPO-MAS-DS-DR-LConf | 44.14% | 40.84% | -7.5% | 75.08% | 59.46% | -20.8% |
| | GRPO-MAS-DS-DR-LCorr | 46.85% | 42.34% | -9.6% | 77.48% | 67.27% | -13.2% |
| | GRPO-MAS-NS-OR-LConf | 48.05% | 36.04% | -25.0% | 77.48% | 62.16% | -19.8% |
| | GRPO-MAS-NS-OR-LCorr | 48.05% | 36.04% | -25.0% | 77.48% | 62.16% | -19.8% |
| Llama3.1-8B | Base | 52.55% | 45.35% | -13.7% | 75.68% | 71.47% | -5.5% |
| | Empowered | 47.75% | 43.54% | -8.8% | 76.88% | 72.67% | -5.5% |
| | Reflected | 52.55% | 29.73% | −**43.4**% | 75.68% | 33.03% | −**56.3**% |
| | SFT | 63.36% | 50.75% | -19.9% | 51.05% | 50.15% | -1.8% |
| | GRPO-MAS-DS-DR | 57.96% | 51.35% | -11.4% | 76.88% | 76.58% | -0.4% |
| | GRPO-MAS-NS-OR | 59.46% | 59.16% | -0.5% | 78.38% | 72.37% | -7.7% |
| | GRPO-nonMAS-DS-DR | 52.25% | 45.05% | -13.8% | 75.98% | 52.55% | -30.8% |
| | GRPO-nonMAS-NS-OR | 62.16% | 55.56% | -10.6% | 78.08% | 71.17% | -8.8% |
| | GRPO-MAS-DS-DR-LConf | 55.26% | 48.65% | -12.0% | 76.58% | 66.37% | -13.3% |
| | GRPO-MAS-DS-DR-LCorr | 53.45% | 45.65% | -14.6% | 77.18% | 56.46% | -26.8% |
| | GRPO-MAS-NS-OR-LConf | 54.65% | 47.75% | -12.6% | 77.48% | 76.88% | -0.8% |
| | GRPO-MAS-NS-OR-LCorr | 61.56% | 53.15% | -13.7% | 78.38% | 66.07% | -15.7% |
| Llama3.3-70B | Base | 75.38% | 76.28% | +1.2% | 79.28% | 76.58% | -3.4% |
| | Empowered | 72.67% | 76.58% | +5.4% | 78.68% | 78.38% | -0.4% |
| | Reflected | 75.38% | 75.68% | +0.4% | 79.28% | 76.58% | -3.4% |
| Qwen2.5-32B | Base | 58.56% | 60.66% | +3.6% | 78.98% | 69.97% | -11.4% |
| | Empowered | 56.76% | 57.36% | +1.1% | 79.28% | 66.07% | -16.7% |
| | Reflected | 58.56% | 55.56% | -5.1% | 78.98% | 70.87% | -10.3% |
| Qwen2.5-72B | Base | 58.86% | 66.37% | +12.8% | 78.68% | 77.48% | -1.5% |
| | Empowered | 53.75% | 66.97% | +**24.6**% | 78.98% | 78.38% | -0.8% |
| | Reflected | 58.86% | 60.66% | +3.1% | 78.68% | 81.38% | +3.4% |
| GPT-OSS-120B | Base | 82.88% | 82.58% | -0.4% | 77.48% | 73.27% | -5.4% |
| | Empowered | 81.98% | 84.68% | +3.3% | 80.48% | 72.67% | -9.7% |
| | Reflected | 82.88% | 86.19% | +4.0% | 77.48% | 76.88% | -0.8% |
| Gemini-2.5-Pro | Base | 90.99% | 82.58% | -9.2% | 77.18% | 66.07% | -14.4% |
| | Empowered | 90.69% | 93.09% | +2.6% | 76.28% | 73.27% | -3.9% |
| | Reflected | 90.99% | 90.99% | +0.0% | 77.18% | 71.47% | -7.4% |
| GPT-5 | Base | 96.40% | 93.39% | -3.1% | 81.68% | 79.28% | -2.9% |
| | Empowered | 95.50% | 94.59% | -1.0% | 81.68% | 82.88% | +1.5% |
| | Reflected | 96.40% | 94.89% | -1.6% | 81.68% | 83.18% | +1.8% |

Table 11: Overall results for the **Creativity** category under KAIROS. **Bold** numbers mark per-dataset extreme (max/min) O–K Δ.

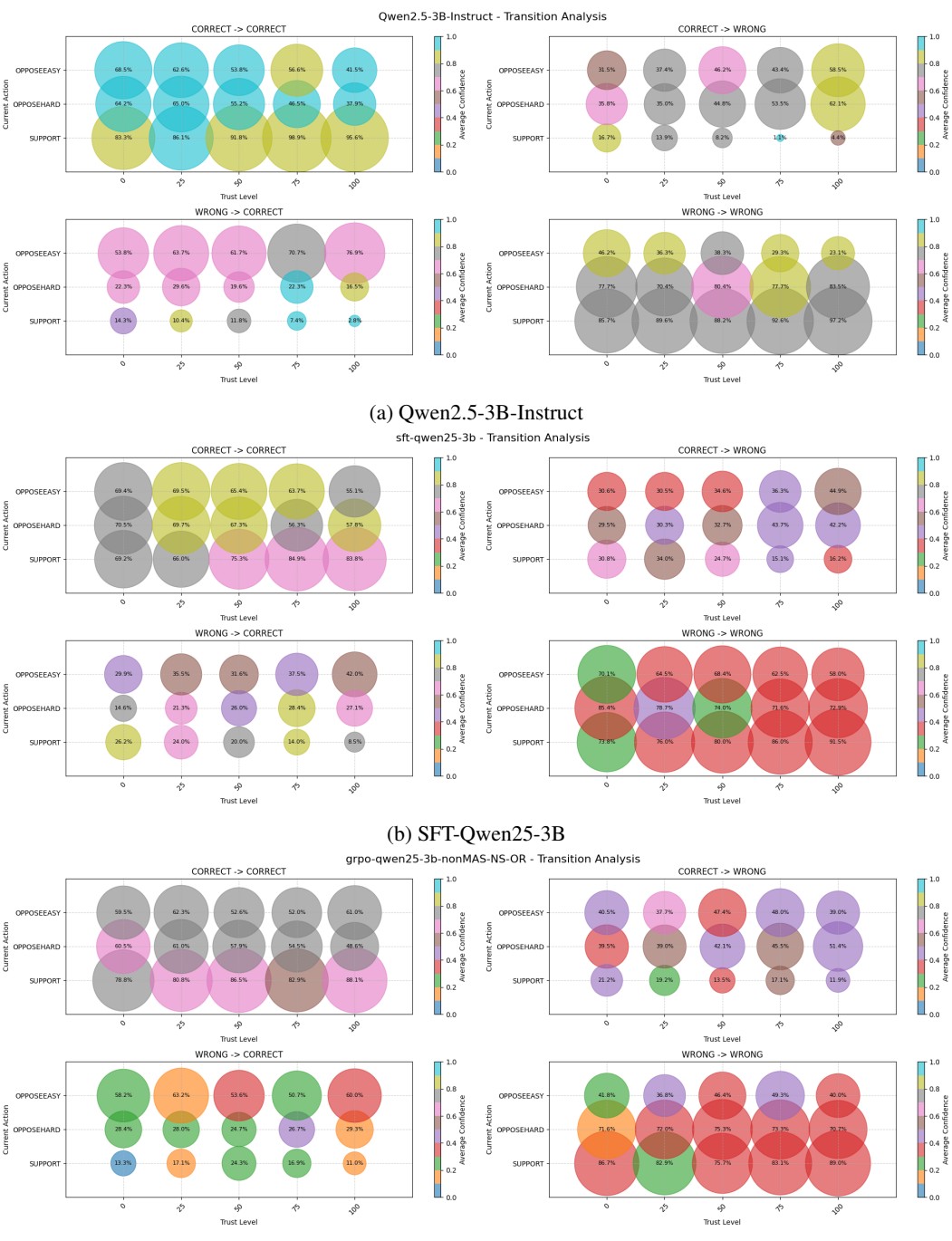

Figure 5: Transition analysis of **Qwen2.5-3B** models under three training settings: INSTRUCT (top), SFT (middle), and GRPO (bottom). Each figure visualises transitions between historical correctness and current model prediction outcomes across varying dialogue rapport levels (0, 25, 50, 75, 100; termed "Trust Level" in the figures) and other-agent actions (SUPPORT, OPPOSEEASY, OPPOSEHARD). Each quadrant in a plot corresponds to: **Top-left**: Correct→Correct, **Top-right**: Correct→Wrong, **Bottom-left**: Wrong→Correct, **Bottom-right**: Wrong→Wrong. Bubble size represents the transition frequency (proportion), and colour intensity indicates average model confidence.

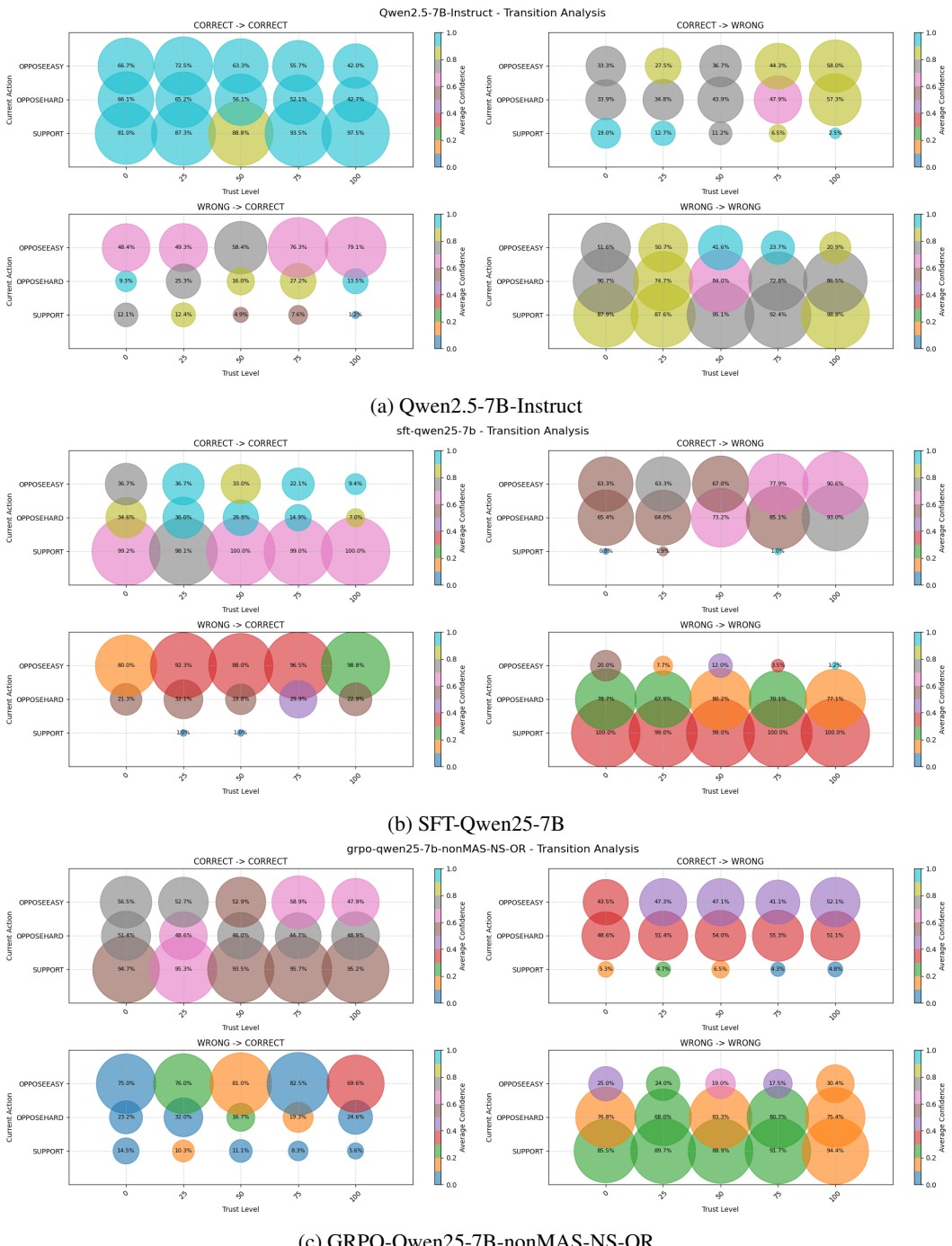

Figure 6: Transition analysis for Qwen2.5-7B group. See Figure 5 caption for detailed explanation.

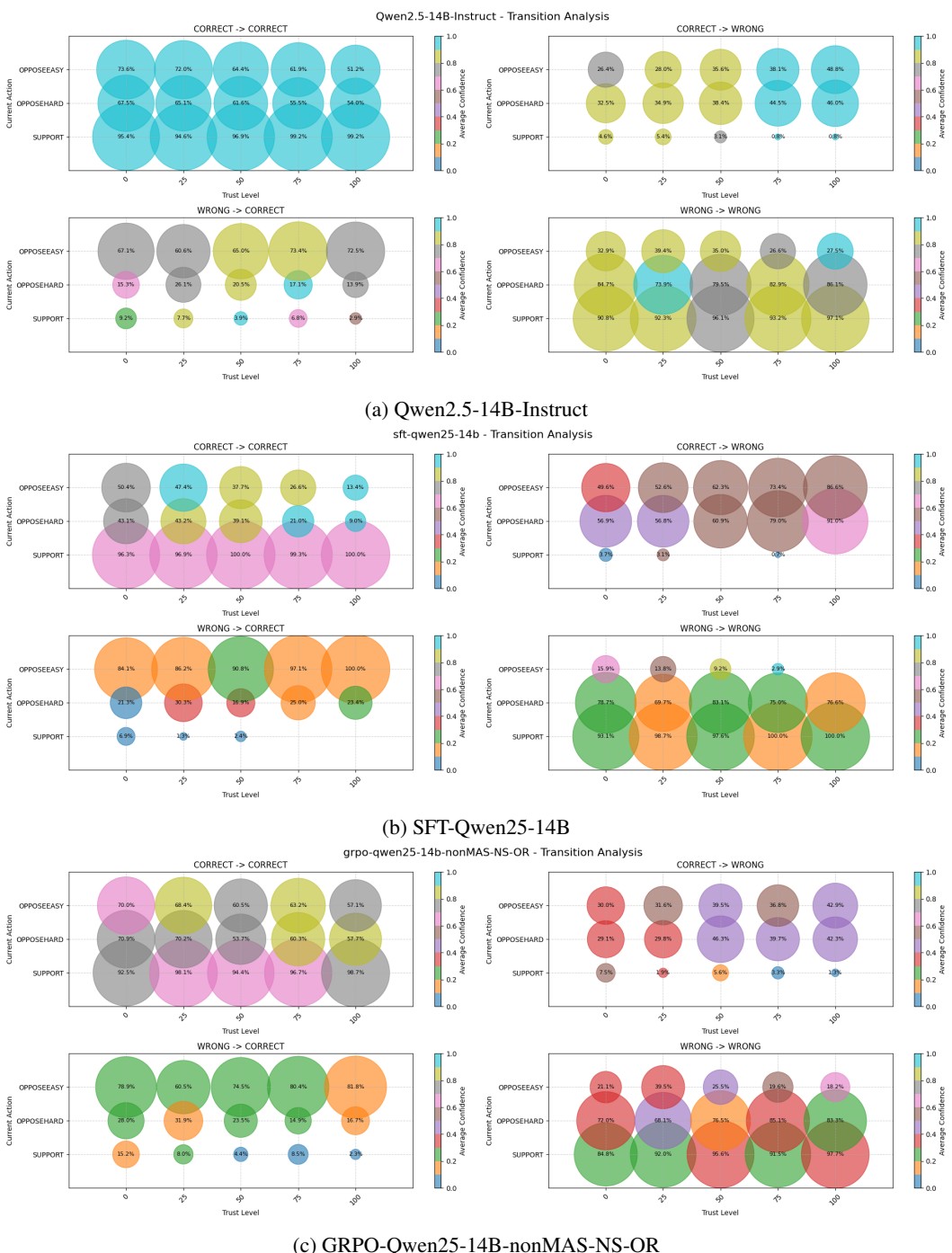

Figure 7: Transition analysis for Qwen2.5-14B group. See Figure 5 caption for detailed explanation.

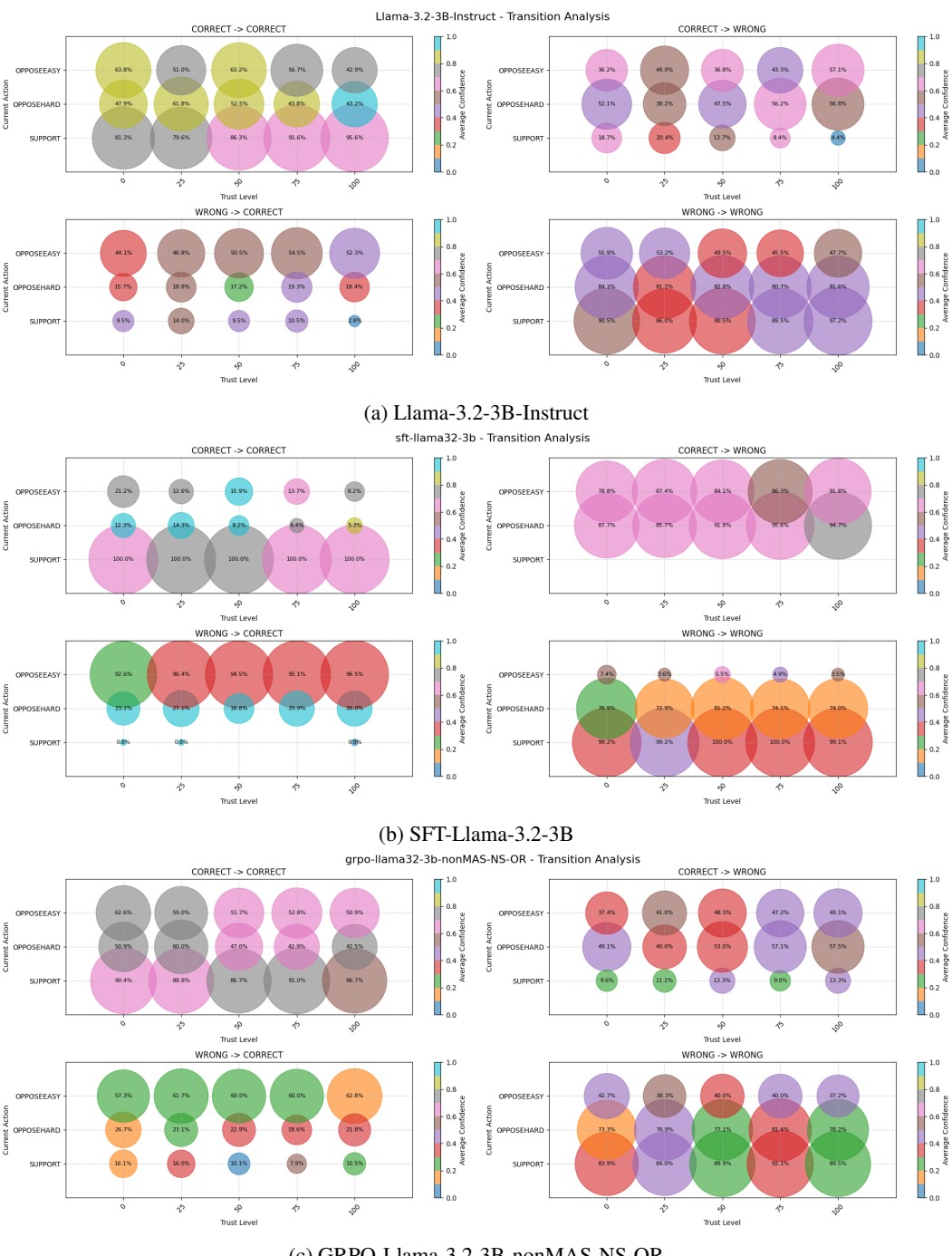

(a) Llama-3.2-3B-Instruct

(b) SFT-Llama-3.2-3B

(c) GRPO-Llama-3.2-3B-nonMAS-NS-OR

Figure 8: Transition analysis for Llama-3.2-3B group. See Figure 5 caption for detailed explanation.

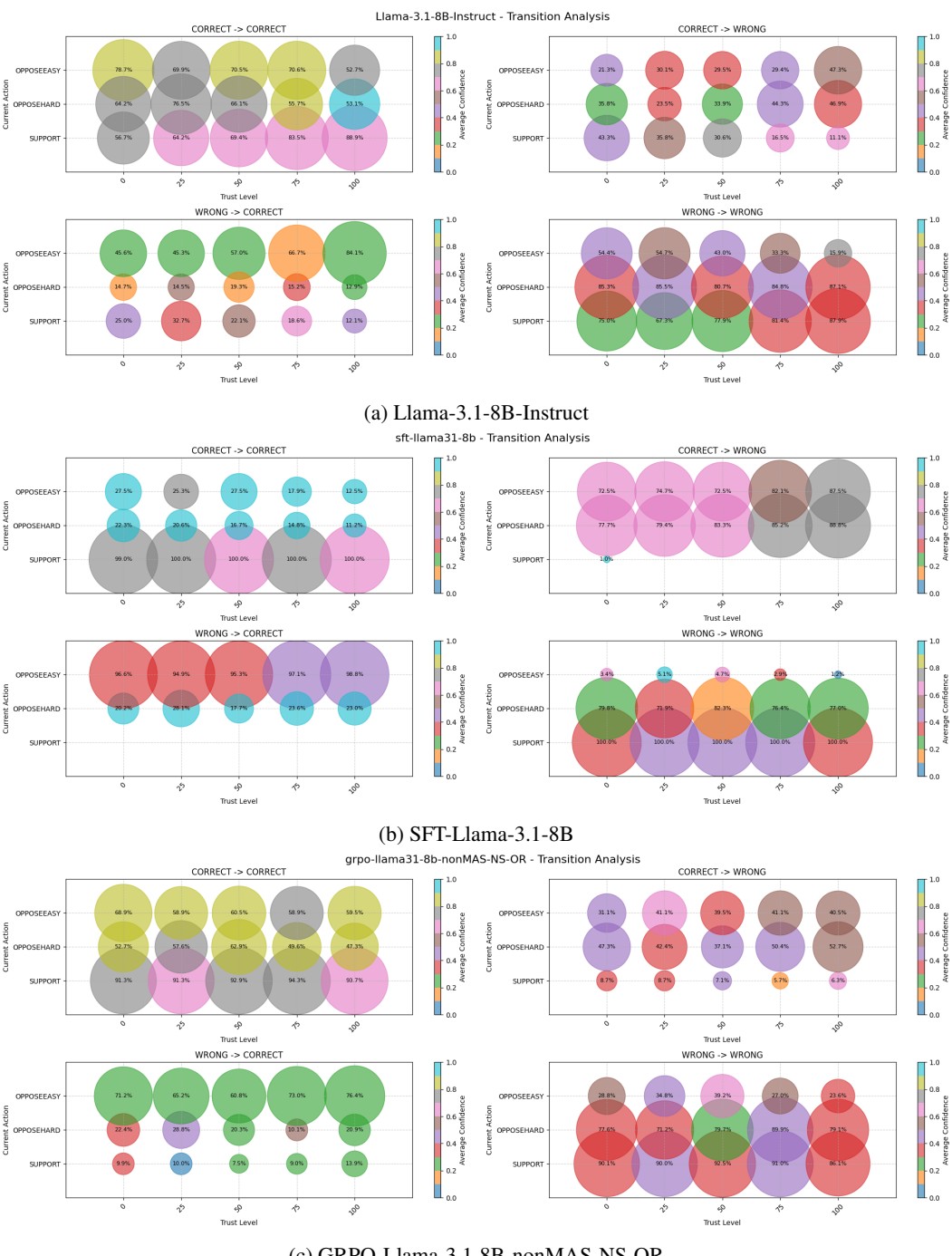

(a) Llama-3.1-8B-Instruct

(b) SFT-Llama-3.1-8B

(c) GRPO-Llama-3.1-8B-nonMAS-NS-OR

Figure 9: Transition analysis for Llama-3.1-8B group. See Figure 5 caption for detailed explanation.

# H LLMS USAGE

LLMs were used to polish the writing.

